# A LAW OF DATA RECONSTRUCTION
# FOR RANDOM FEATURES (AND BEYOND)

**Leonardo Iurada**[*†], **Simone Bombari**[*‡], **Tatiana Tommasi**[†], **Marco Mondelli**[‡]
[†] Politecnico di Torino, Italy      [‡] Institute of Science and Technology Austria (ISTA)

## ABSTRACT

Large-scale deep learning models are known to *memorize* parts of the training set. In machine learning theory, memorization is often framed as interpolation or label fitting, and classical results show that this can be achieved when the number of parameters $p$ in the model is larger than the number of training samples $n$. In this work, we consider memorization from the perspective of *data reconstruction*, demonstrating that this can be achieved when $p$ is larger than $dn$, where $d$ is the dimensionality of the data. More specifically, we show that, in the random features model, when $p \gg dn$, the subspace spanned by the training samples in feature space gives sufficient information to identify the individual samples in input space. Our analysis suggests an optimization method to reconstruct the dataset from the model parameters, and we demonstrate that this method performs well on various architectures (random features, two-layer fully-connected and deep residual networks). Our results reveal a *law of data reconstruction*, according to which the entire training dataset can be recovered as $p$ exceeds the threshold $dn$.

## 1 INTRODUCTION

*How many parameters does a neural network need to memorize the training data?*

The answer to this question depends on what one means by *memorization*, a term used with different purposes in the machine learning literature. Informally speaking, it captures the phenomenon of models storing the information of individual training samples, as opposed to learning the statistical patterns in the data. Thus, on the one hand, it is used to describe the phenomenon of label fitting (Zhang et al., 2017) or forms of leave-one-out output stability (Feldman, 2020; Feldman & Zhang, 2020). On the other hand, memorization is associated with *reconstructing* parts of the *training set* through knowledge of the model parameters (Carlini et al., 2023b; Schwarzschild et al., 2024; Cooper & Grimmelmann, 2024). Notably, this reconstruction is possible in modern foundation models (Carlini et al., 2021; Nasr et al., 2025; Cooper et al., 2025), with consequences in terms of privacy concerns and copyright infringement (Tramèr et al., 2024; Cooper & Grimmelmann, 2024).

Understanding how many parameters a neural network needs to interpolate the dataset (or, equivalently, memorize the labels) is a classical problem (Cover, 1965), with more modern literature showing that interpolation occurs as soon as the number of model parameters $p$ exceeds the number of training samples $n$ (Soltanolkotabi et al., 2018; Montanari & Zhong, 2022; Bombari et al., 2022). An intuition for the phenomenon is that solving $n$ equations (given by the training samples) generally requires $p \geq n$ degrees of freedom (given by the model parameters). In contrast, for the problem of reconstructing the training dataset, the situation is less clear. Empirical work has observed that this task becomes easier as the model gets larger (Haim et al., 2022; Carlini et al., 2023b). However, theoretical work has taken different perspectives, focusing *e.g.* on impossibility results for differentially private training (Balle et al., 2022), on reconstructing the data from the model gradients (Wang et al., 2023) or on models with an infinite number of parameters (Loo et al., 2024). To the best of our knowledge, there is no theoretical result connecting feasibility of data reconstruction and model size.

To address this gap, we propose a *law of data reconstruction*, giving a threshold for which reconstruction becomes possible. We consider *random features* (RF) regression, where the model is

---

[*]Joint first authorship, equal contribution
[†]Emails: {`leonardo.iurada, tatiana.tommasi`}`@polito.it`
[‡]Emails: {`simone.bombari, marco.mondelli`}`@ist.ac.at`

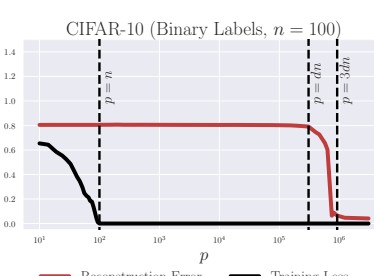
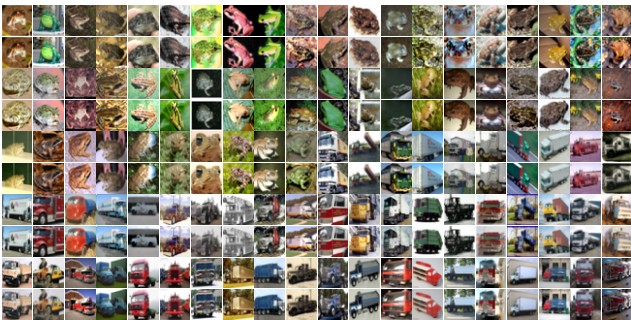

Figure 1: **Thresholds for label fitting and data reconstruction in the random features model.** (Left) We consider RF regression with ReLU activation on binary labels (*frogs vs. trucks*) with $n = 100$ images (50 examples per class) from CIFAR-10 ($d = 3072$). We report the mean for both the reconstruction error (in red, defined in (12)) and the training error (in black, mean squared error), as the number of parameters $p$ increases. Statistics are computed across 4 distinct random seeds, and the standard deviation across seeds is very small (the confidence interval at one standard deviation is reported in the plot as shaded area, but it is imperceptible). For $p \geq n$, training labels are memorized, while reconstruction is feasible when $p$ becomes larger than $dn$. (Right) Results of the reconstruction when $p = 10dn$. Odd rows report the ground truth images, while even rows the reconstructed ones which are all visually very similar.

$f_{\mathrm{RF}}(x, \theta) = \varphi(x)^\top \theta$, see Eq. (1). Here, $x$ is the $d$-dimensional input, $\varphi(x)$ the corresponding $p$-dimensional feature vector, and $\theta$ the $p$-dimensional model parameter vector obtained by running gradient descent on the square loss over $n$ samples. Our paper gives theoretical and empirical evidence that:

$$\text{all } \textit{the training data can be reconstructed when } p \gg dn.$$

An intuition for the phenomenon is that solving $dn$ equations (given by each dimension of each training sample) generally requires $p \geq dn$ degrees of freedom (given by the model parameters). Our analysis also suggests an optimization algorithm to reconstruct the training dataset and, as a proof of concept, we discuss the results of the designed method on CIFAR-10. We train a family of RF models using the square loss on $n = 100$ training samples. The left panel of Figure 1 shows the training loss in black and the *reconstruction error* in red (formally defined in Eq. (12)), as functions of the number of parameters $p$. As expected, the training error converges to 0 after the interpolation threshold $p = n$, where labels are memorized. In contrast, the reconstruction error drops only when $p$ is of order $dn$. In the right panel of Figure 1, we display training images (odd rows) and reconstructed ones (even rows) for $p = 10dn$, showing that the whole dataset is recovered successfully. Our contributions are summarized below:

- In Theorem 1, we consider a set of $n$ points $\hat{x}_1, \ldots, \hat{x}_n \in \mathbb{R}^d$, and prove that when $p \gg dn$, if for every training sample $x_i$ it holds $\varphi(x_i) \in \mathrm{span}\{\{\varphi(\hat{x}_j)\}_{j=1}^n\}$, then all the $\hat{x}_j$-s *must be close* to one of the original training data.

- As the previous result does not exclude the possibility of duplicates within the $\hat{x}_j$-s, in Theorem 2 we prove that the $\hat{x}_j$-s *must be distinct*, focusing on the case $n = 2$ for simplicity. Taken together, Theorem 1 and 2 show that, when $p \gg dn$, the entire training set can be reconstructed given knowledge of the subspace $\mathrm{span}\{\{\varphi(x_i)\}_{i=1}^n\}$. In fact, having access to $\mathrm{span}\{\{\varphi(x_i)\}_{i=1}^n\}$, one can then look for $\hat{x}_1, \ldots, \hat{x}_n$ s.t. $v \in \mathrm{span}\{\{\varphi(\hat{x}_j)\}_{j=1}^n\}$ for any $v \in \mathrm{span}\{\{\varphi(x_j)\}_{j=1}^n\}$.

- In practice, we have access to the vector of trained parameters $\theta^*$, which is in $\mathrm{span}\{\{\varphi(x_i)\}_{i=1}^n\}$, see Eq. (2). This motivates considering the *reconstruction loss* $\|P_{\hat{\Phi}}^\perp \theta^*\|_2^2$, where $P_{\hat{\Phi}}$ is the projector on $\mathrm{span}\{\{\varphi(\hat{x}_j)\}_{i=1}^n\}$: if $\|P_{\hat{\Phi}}^\perp \theta^*\|_2^2 = 0$, then $\theta^* \in \mathrm{span}\{\{\varphi(\hat{x}_j)\}_{j=1}^n\}$. We empirically show that optimizing this loss over the $\hat{x}_j$-s via gradient descent leads to the reconstruction of the training dataset, when $p \gg dn$. Notably, this procedure for data reconstruction is not limited to the RF model, but it performs well also for two-layer and deep residual networks, see Figures 6-7.

We finally remark that the scaling $p \gg dn$ was previously considered in the context of adversarial robustness: prior work proved that the condition $p \gg dn$ is both necessary (Bubeck et al., 2021; Bubeck & Sellke, 2021) and, in some settings, sufficient (Bombari et al., 2023) for *smooth* label interpolation. This suggests an inherent connection between the adversarial robustness of the model and the ability to reconstruct training data from knowledge of its parameters.

## 2 RELATED WORK

**Over-parameterization and memorization.** The problem of label memorization and how it relates to the number of parameters in a neural network dates back to seminal results (Cover, 1965; Baum, 1988). More recently, a line of work aimed at showing that gradient descent converges to 0 training loss (i.e., the model memorizes the labels) for networks with progressively smaller over-parameterization (Allen-Zhu et al., 2019; Du et al., 2019b;a; Oymak & Soltanolkotabi, 2020; Nguyen & Mondelli, 2020; Nguyen, 2021; Nguyen et al., 2021; Bombari et al., 2022), with Bombari et al. (2022) proving that $p \gg n$ parameters are sufficient to fit any set of labels in deep neural networks. Under a separability assumption, it has also been shown that $n$ arbitrary training points can be perfectly classified by a neural network with $p \gg \sqrt{n}$ parameters (Vardi et al., 2022). This, in the context of regression, implies that $p \gg \log(1/\epsilon)\sqrt{n}$ is sufficient to guarantee an error of at most $\epsilon$. In contrast, input data memorization and reconstruction is less explored from a theoretical standpoint. Balle et al. (2022) proved impossibility results if the model is trained with differential privacy (Dwork et al., 2006), and showed that in generalized linear models it is possible to reconstruct an individual data provided all other samples are known. Loo et al. (2024) considered networks with an infinite number of parameters, Smorodinsky et al. (2024) focused on partial uni-dimensional ($d = 1$) training set reconstruction, and Wang et al. (2023) assumed knowledge of model gradients, requiring $p \gg d^2$ parameters with $n$ at most of order $d^{1/4}$. The regime $p \gg dn$ was heuristically identified by Haim et al. (2022) as enabling successful data reconstruction, while (Brown et al., 2021; Feldman et al., 2025) studied its role as a requirement for learning certain tasks, rather than for data reconstruction.

**Data reconstruction.** Recent studies have shown that training data can be extracted from generative models by carefully prompting their generation routines (Carlini et al., 2023a; Zhang et al., 2023; Nasr et al., 2025; Cooper et al., 2025). More generally, data reconstruction is a broader task, not limited to generative models, that aims to recover the training set directly from the learned model's parameters (Cooper & Grimmelmann, 2024). A possible approach is via model inversion attacks, which optimize over the input space to raise a class score, although this method generally recovers class prototypes rather than specific training samples (Fredrikson et al., 2015; Mahendran & Vedaldi, 2015). Another strategy relies on the implicit bias of gradient descent for homogeneous networks (Lyu & Li, 2020; Ji & Telgarsky, 2020): Haim et al. (2022) proposed a reconstruction objective motivated by the observation that gradient descent converges to points satisfying the KKT conditions of a max-margin problem, and empirically showed that training samples on the classification margin can be recovered. This method was then extended by Buzaglo et al. (2023); Oz et al. (2024) to the multiclass setting with general losses, and to larger scale pretrained models. Loo et al. (2024) adapted the idea to neural networks trained with the square loss in the linear (or NTK) regime (Jacot et al., 2018; Lee et al., 2019). This leads to an optimization method for data reconstruction which has similarities with ours (compare Eq. (7) in (Loo et al., 2024) with Eq. (11) in this work).

**Random features (RF) regression.** The RF model (Rahimi & Recht, 2007) can be regarded as a two-layer network with random first layer weights. Its popularity stems from its mathematical tractability and, in contrast with linear regression where number of parameters equals input dimension, it captures the statistical effects of over-parameterization, as the number of parameters $p$ scales independently from $d$ and $n$. Mei & Montanari (2022) characterized the test loss of random features, showing that it displays double descent (Belkin et al., 2019). Furthermore, the RF model has been used to understand various phenomena such as feature learning (Ba et al., 2022; Damian et al., 2022; Moniri et al., 2023), robustness under adversarial attacks (Dohmatob & Bietti, 2022; Bombari et al., 2023; Hassani & Javanmard, 2024), distribution shift (Tripuraneni et al., 2021; Lee et al., 2023; Bombari & Mondelli, 2025b), and scaling laws (Defilippis et al., 2024; Paquette et al., 2024).

## 3 PROBLEM SETUP

**Notation.** All complexity notations $\omega, \Omega, \Theta, O, o$ are understood for sufficiently large input dimension $d$ and number of parameters $p$. Given a positive number $k$, $[k]$ denotes the set of positive numbers from 1 to $k$. Given a vector $v$, $\|v\|_2$ denotes its Euclidean norm. Given a matrix $A \in \mathbb{R}^{m \times n}$, we denote by $P_A \in \mathbb{R}^{n \times n}$ the projector over $\mathrm{span}\{\mathrm{rows}(A)\}$. We say that an event holds with overwhelming probability if it holds with probability at least $1 - e^{-\omega(\log d)}$.

Let $(X, Y)$ be a labeled training dataset, where $X = [x_1, \ldots, x_n]^\top \in \mathbb{R}^{n \times d}$ contains the training input data (sampled i.i.d. from a distribution $\mathcal{P}_X$) on its rows and $Y = (y_1, \ldots, y_n) \in \mathbb{R}^n$ contains the corresponding labels. We define the *random features* (RF) model as

$$f_{\text{RF}}(x, \theta) = \phi(Vx)^\top \theta = \varphi(x)^\top \theta, \tag{1}$$

where $\varphi(x) \in \mathbb{R}^p$ is the feature vector associated to the input $x \in \mathbb{R}^d$, and $\theta \in \mathbb{R}^p$ are trainable parameters. The random features matrix $V \in \mathbb{R}^{p \times d}$ is s.t. $V_{i,j} \sim_{\text{i.i.d.}} \mathcal{N}(0, 1/d)$, and $\phi : \mathbb{R} \to \mathbb{R}$ is a non-linearity applied component-wise. We note that $f_{\text{RF}}(x, \theta)$ is a generalized linear model, and it can be regarded as a two-layer fully-connected neural network, where only the second layer is trained. We consider the supervised learning setup where a quadratic training loss (without regularization) is minimized via gradient descent. When the learning rate is sufficiently small, gradient descent converges to the interpolator which is the closest in $\ell_2$ norm to the initialization (see Equation (33) in (Bartlett et al., 2021)), which we consider equal to 0 for simplicity. Then, denoting with $\Phi := [\varphi(x_1), \dots, \varphi(x_n)]^\top \in \mathbb{R}^{n \times p}$ the feature matrix, the trained parameters are

$$\theta^* = \Phi^+ Y, \tag{2}$$

where $\Phi^+$ is the Moore-Penrose inverse of $\Phi$.

**Assumption 1** (Data distribution)**.** *The training samples $\{x_1, \dots, x_n\}$ are $n$ i.i.d. samples from the sub-Gaussian distribution $\mathcal{P}_X$, with $\|x\|_{\psi_2} = O(1)$, and such that $\|x\|_2 = \sqrt{d}$.*

This assumption requires the data to have well-behaved tails (see the definition of the sub-Gaussian norm $\|\cdot\|_{\psi_2}$ in Eq. (13) in Appendix A for further details), and allows for badly conditioned distributions. This hypothesis is commonly used in the related literature, and it includes *e.g.* Gaussian data, data respecting Lipschitz concentration (Bubeck & Sellke, 2021; Bombari et al., 2023; von Berg et al., 2025), and data uniform on the sphere (Mei & Montanari, 2022; Hu et al., 2024). The normalization $\|x\|_2 = \sqrt{d}$ is chosen for technical convenience, and this scaling of the norm (combined with the scaling of the random features matrix $V$) guarantees that the pre-activations of the model (*i.e.*, the entries of $Vx$) are of constant order.

**Assumption 2** (Activation function)**.** *The activation function $\phi : \mathbb{R} \to \mathbb{R}$ is a non-linear, Lipschitz continuous function such that its derivative is also Lipschitz. Letting $\mu_l$ denote the $l$-th Hermite coefficient of $\phi$, we further assume that $\mu_0 = \mu_2 = 0$, $\mu_1 \neq 0$, and that there exist two non-zero Hermite coefficients of order $\geq 3$ with different parity.*

This assumption is motivated by theoretical convenience, and we expect our results to hold for a wider set of activations (as in (Mei et al., 2022)) after a more involved analysis. Except for the last condition on the parity of high-order Hermite coefficients, Assumption 2 resembles the setting considered by Hu & Lu (2022), and it covers a wide family of odd activations, including $\tanh$. The parity of high-order Hermite coefficients is used to reconstruct the correct sign of the training data, and it appears to be necessary for that, see Remark 1 for details.

**Assumption 3** (Over-parameterization)**.** *We consider a data dimensionality regime where $n = O(d)$, and an over-parameterized model with*

$$p = \omega \left( nd \log^2 d \right). \tag{3}$$

To guarantee that the RF model interpolates the data, it suffices that $p \gg n$ (Mei et al., 2022; Wang & Zhu, 2024). The stronger over-parameterization requirement in Eq. (3) was shown to be both necessary (Bubeck & Sellke, 2021) and, in some settings, sufficient (Bombari et al., 2023) to achieve *smooth* interpolation. We focus on the regime $n = O(d)$, which includes the popular proportional regime $n = \Theta(d)$ as well as the regime where $n$ is a fixed constant (independent of $d, p$). We expect our results to be generalizable also to $n \gg d$ (Mei et al., 2022; Hu et al., 2024; Pandit et al., 2024).

## 4 MAIN RESULTS

In this section, we present our main theoretical results, pinpointing $p \gg dn$ as the over-parameterization threshold such that data reconstruction can take place given knowledge of the subspace of the training samples in feature space.

More formally, the goal of data reconstruction is to exhibit a matrix $\hat{X} \in \mathbb{R}^{n \times d}$ such that its rows $\hat{x}_j \in \mathbb{R}^d$ are close to the rows of the original training data matrix $X$. When $\|\hat{x}_j\|_2 = \|x_i\|_2 = \sqrt{d}$, a reconstructed sample $\hat{x}_j$ can be considered close to a training sample $x_i$ if $\|\hat{x}_j - x_i\|_2 = o(\sqrt{d})$. Let us also define $\hat{\Phi} = [\varphi(\hat{x}_1), \dots, \varphi(\hat{x}_n)] \in \mathbb{R}^{n \times p}$ as the feature matrix of the reconstructed data. Then, the result below gives sufficient conditions for all rows of $\hat{X}$ to be close to training samples.

**Theorem 1.** *Let Assumptions 1, 2, and 3 hold. Let $\hat{X} \in \mathbb{R}^{n \times d}$ be such that its rows satisfy $\|\hat{x}_i\|_2 = \sqrt{d}$, and for every $i \in [n]$, $\varphi(x_i) \in \mathrm{span}\{\mathrm{rows}(\hat{\Phi})\}$. Then, with overwhelming probability, for any $\hat{\imath} \in [n]$, there exists $i \in [n]$ such that*

$$\|\hat{x}_{\hat{\imath}} - x_i\|_2 = o(\sqrt{d}). \tag{4}$$

In words, Theorem 1 states that, if the random features of training samples are spanned by the random features of a matrix $\hat{X}$, the rows of $\hat{X}$ *must* be close (in input space) to the original training samples. Geometrically, this result proves that, when $p \gg dn$, in order to span the subspace generated by the training data features, one has to consider approximately the same vectors, as there is no solution to this problem obtained by non-degenerate linear combinations. In contrast with (Loo et al., 2024) which tackles the case $p \to \infty$, our main contribution is to pinpoint the threshold $p \gg dn$ constituting a sufficient amount of over-parameterization to reconstruct the data. We also highlight that, for the claim of Theorem 1 to hold, assumptions on the activation are necessary: if $\phi$ is linear, picking $\hat{x}_i = \sqrt{d}(x_i + x_{i+1})/\|x_i + x_{i+1}\|_2$ ($i \in [n-1]$) and $\hat{x}_n = \sqrt{d}(x_n - x_1)/\|x_n - x_1\|_2$ gives a counterexample to Eq. (4). The proof of Theorem 1 is deferred to Appendix B, and a sketch is below.

*Proof sketch.* Prior results on the RF kernel concentration (Mei et al., 2022; Wang & Zhu, 2024) guarantee that, for $p \gg n$, the smallest eigenvalue $\lambda_{\min}(\Phi\Phi^\top)$ is bounded away from 0. Thus, the rows of $\Phi$ are linearly independent (they span a sub-space of dimension *exactly* $n$). Then, as the $n$ row vectors of $\hat{\Phi}$ span all rows of $\Phi$ (by hypothesis), $\mathrm{span}\{\mathrm{rows}(\Phi)\} = \mathrm{span}\{\mathrm{rows}(\hat{\Phi})\}$. This in turn gives that, if $\hat{x} \in \mathbb{R}^d$ is a generic row of $\hat{X}$, then $\varphi(\hat{x}) \in \mathrm{span}\{\mathrm{rows}(\Phi)\}$, i.e.,

$$\varphi(\hat{x}) = \sum_{i=1}^{n} a_i \varphi(x_i). \tag{5}$$

As argued above, the result cannot hold for linear activations, so let us focus on the non-linear component $\tilde{\varphi}(x_i) = \phi(Vx_i) - \mu_1 Vx_i$ in the Hermite basis. Taking the inner product of both sides of (5) with $\tilde{\varphi}(x_i)$ and with $\tilde{\varphi}(\hat{x})$ yields, with overwhelming probability,

$$\left| \tilde{\varphi}(x_i)^\top \tilde{\varphi}(\hat{x}) - \tilde{\mu}^2 a_i \right| = \tilde{O}\left(\sqrt{\frac{dn}{p}}\right) + o(1), \qquad \left| \|a\|_2^2 - 1 \right| = \tilde{O}\left(\sqrt{\frac{dn}{p}}\right) + o(1), \tag{6}$$

where $a \in \mathbb{R}^n$ is defined as the vector containing $a_i$ in its $i$-th entry, and $\tilde{\mu}^2$ is the sum of the squares of the Hermite coefficients of $\phi$ of order at least 3. The two results in Eq. (6) are formalized in Lemmas B.2-B.3, and they rely on concentration arguments on the random features $V$, which have to hold *uniformly* over any $\hat{x} \in \sqrt{d}\mathbb{S}^{d-1}$. To obtain such uniform concentration, we rely on an $\epsilon$-net argument which crucially uses the condition $p \gg dn$.

Next, in Lemma B.4, we show that, with overwhelming probability,

$$\text{if} \quad C = \max_i \left| \frac{x_i^\top \hat{x}}{d} \right|, \qquad \text{then} \quad \left\| \frac{(X\hat{x})^{\circ l}}{d^l} \right\|_2 \le C^{l-1} + o(1), \tag{7}$$

uniformly for every $l \ge 2$. Combining Eq. (6)-(7) gives that $|C - 1| = o(1)$, i.e., there exists a single training sample $x_j$ aligned with $\hat{x}$. To resolve the ambiguity in the sign, we use that there exist two non-zero Hermite coefficients of order $\ge 3$ with different parity, which concludes the argument. $\square$

**Remark 1** (Sign ambiguity). *The existence of two non-zero Hermite coefficients with different parity is a necessary condition to recover the sign of the training samples. In fact, if $\phi$ is either even or odd, the problem is under-determined in terms of the sign of the $\hat{x}_i$-s, as $\mathrm{span}\{\mathrm{rows}(\hat{\Phi})\}$ does not depend on them. Remarkably, this effect is also evident in numerical experiments optimizing the reconstruction loss defined in Eq. (11): Figure 4 considers ReLU activation (which violates the last condition of Assumption 2, as its Hermite coefficients $\mu_{2l+1} = 0$ for all $l > 1$) showing that negatives of training samples may be reconstructed.*

Theorem 1 guarantees that all the rows of $\hat{X}$ are close to the training samples. However, it may still happen that multiple rows of $\hat{X}$ are close to the same sample, leaving part of the training dataset not reconstructed. This gap is approached by our next result which focuses on the case $n = 2$.

**Theorem 2.** *Let Assumptions 1, 2, and 3 hold. Let $n = 2$ and $\hat{X} \in \mathbb{R}^{2 \times d}$ be such that its rows satisfy $\|\hat{x}_i\|_2 = \sqrt{d}$, and for every $i \in \{1, 2\}$, $\varphi(x_i) \in \mathrm{span}\{\mathrm{rows}(\hat{\Phi})\}$. Then, with overwhelming probability, for any $i \in \{1, 2\}$, there exists $\hat{\imath} \in \{1, 2\}$ such that*

$$\|\hat{x}_{\hat{\imath}} - x_i\|_2 = o(\sqrt{d}). \tag{8}$$

In words, Theorem 2 shows that *all* training samples are in fact reconstructed, ruling out the possibility of $\hat{X}$ containing repetitions. The proof is deferred to Appendix C and a sketch is below.

*Proof sketch.* By contradiction, suppose there exist two vectors $\epsilon_1, \epsilon_2 \in \mathbb{R}^d$ such that

$$\|\epsilon_1\|_2, \|\epsilon_2\|_2 = o(\sqrt{d}), \qquad \varphi(x_2) = a_1 \varphi(x_1 + \epsilon_1) + a_2 \varphi(x_1 + \epsilon_2),$$

for some real values $a_1$ and $a_2$. A Taylor expansion of $\varphi(x_1 + \epsilon_2)$ around $x_1 + \epsilon_1$ gives

$$
\begin{aligned}
\varphi(x_2) = {} & (a_1 + a_2)\varphi(x_1 + \epsilon_1) + a_2\, \phi'(V(x_1 + \epsilon_1)) \circ (V(\epsilon_2 - \epsilon_1)) \\
& + a_2 \left( (\varphi(x_1 + \epsilon_2) - \varphi(x_1 + \epsilon_1)) - \phi'(V(x_1 + \epsilon_1)) \circ (V(\epsilon_2 - \epsilon_1)) \right),
\end{aligned}
\tag{9}
$$

where $\circ$ denotes the component-wise product of two vectors. First, we take the inner product of both sides of Eq. (9) with $V x_1$, which yields

$$|a_1 + a_2| = O\left( \frac{|a_2|\, \|\epsilon_2 - \epsilon_1\|_2}{\sqrt{d}} \right) + o(1),$$

with overwhelming probability (see Lemma C.5). Then, we take the inner product with $V(\epsilon_2 - \epsilon_1)$, and show that the first and third term of the RHS of Eq. (9) are negligible with respect to the second one (see Lemmas C.3 and C.4). An upper bound on the LHS of Eq. (9) based on Cauchy-Schwartz inequality is then enough to show in Lemma C.6 that

$$|a_2| = O\left( \frac{\sqrt{d}}{\|\epsilon_2 - \epsilon_1\|_2} \right), \qquad |a_1 + a_2| = O(1),$$

with overwhelming probability. The argument in Lemma C.3 relies on a concentration result on the sum of independent random variables with sub-exponential norm much larger than their standard deviation (see Lemma C.2). This is obtained via a Bernstein-type bound combined with an $\epsilon$-net argument as, similarly to Theorem 1, we need a *uniform* control over any choice of $\epsilon_1, \epsilon_2$. Finally, by taking the inner product of the two sides of Eq. (9) with $\tilde{\varphi}(x_2)$, we obtain

$$\tilde{\varphi}(x_2)^\top \varphi(x_2) \leq \left| (a_2 + a_2)\tilde{\varphi}(x_2)^\top \varphi(x_1 + \epsilon_1) \right| + \left| a_2 \tilde{\varphi}(x_2)^\top \left( \varphi(x_1 + \epsilon_2) - \varphi(x_1 + \epsilon_1) \right) \right|. \tag{10}$$

Now, the LHS of Eq. (10) is $\Theta(p)$ since $\phi$ is non-linear; the first term in the RHS of Eq. (10) is $o(p)$ as $x_1$ and $x_2$ are roughly orthogonal with overwhelming probability; and the last term in the RHS of Eq. (10) is also $o(p)$ via generalized Stein's lemma (see Lemma C.7). This gives a contradiction. $\quad\square$

**Remark 2** (Technical challenge for $n \geq 3$). *We note that, already when $n = 3$, the presence of duplicates is either given by the "triplet" $\hat{x}_1, \hat{x}_2, \hat{x}_3$ all similar to each other, or by a pair of duplicates $\hat{x}_1, \hat{x}_2$, with a different $\hat{x}_3$. The constructive approach used in the proof of Theorem 2 would require us to consider the two cases separately, and the amount of cases increases combinatorially with $n$. Nevertheless, we suspect the idea that the span of duplicate samples cannot contain higher order terms of the left-out samples ($\tilde{\varphi}(x_2)$ in Eq. (10)) to carry over to a general $n$, as long as $p \gg dn$.*

**From the theory to a reconstruction algorithm.** Our theoretical analysis shows that, under sufficient over-parameterization ($p \gg dn$), the matrix $\hat{X}$ successfully reconstructs the training dataset when $\|P_{\hat{\Phi}}^{\perp} \varphi(x_i)\|_2 = 0$ for every $i \in [n]$, where $P_{\hat{\Phi}}$ denotes the projector on $\mathrm{span}\{\mathrm{rows}(\hat{\Phi})\}$. In practice, we only have access to the trained model $\theta^*$, $V$ and the activation $\phi$. Recall $\theta^* = \Phi^+ Y \in \mathrm{span}\{\mathrm{rows}(\Phi)\}$, so $\theta^*$ is a linear combination of $\{\varphi(x_i)\}_{i=1}^n$, which suggests to solve the problem:

$$\hat{X}^* = \underset{\hat{X}\,:\,\|\hat{x}_i\|_2 = \sqrt{d}}{\arg\min} \mathcal{L}(\hat{X}), \quad \mathcal{L}(\hat{X}) = \left\| P_{\hat{\Phi}}^{\perp} \theta^* \right\|_2^2. \tag{11}$$

Importantly, enforcing that $\theta^*$ lies in the span of the reconstructed features ($\mathcal{L}(\hat{X}) = 0$) does not immediately imply that $\varphi(x_i) \in \mathrm{span}\{\mathrm{rows}(\hat{\Phi})\}$ for all $i$ (as the implication only goes in the other direction). In Figure 2, we numerically minimize $\mathcal{L}(\hat{X})$ and check whether the vectors $\varphi(x_i)$ approximately lie in the subspace $\mathrm{span}\{\mathrm{rows}(\hat{\Phi})\}$. To do so, we calculate the average per-feature orthogonal residual, *i.e.* the average over $i \in [n]$ of $\|P_{\hat{\Phi}}^{\perp} \varphi(x_i)\|_2 / \sqrt{p}$, where $P_{\hat{\Phi}}^{\perp} = I - P_{\hat{\Phi}}$ projects onto the orthogonal complement of $\mathrm{span}\{\mathrm{rows}(\hat{\Phi})\}$. This quantity equals 0 if and only if every $\varphi(x_i)$ lies in $\mathrm{span}\{\mathrm{rows}(\hat{\Phi})\}$. The normalization by $\sqrt{p}$ makes $r(\hat{\Phi}) := \sum_{i=1}^n \|P_{\hat{\Phi}}^{\perp} \varphi(x_i)\|_2 / (n\sqrt{p})$ of order 1 so that values of $r(\hat{\Phi}) \ll 1$ indicate numerically negligible residuals (*i.e.*, effective span inclusion).

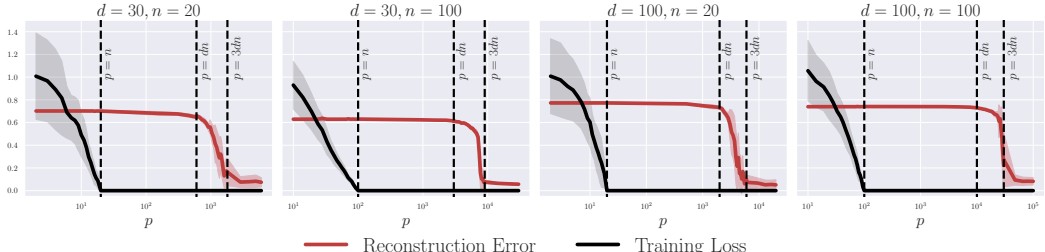

Figure 3: **Thresholds for label fitting and data reconstruction when training on i.i.d. data uniformly drawn from the $d$-dimensional sphere.** We consider RF regression with ReLU activation, fitting a noisy linear model. We report mean (solid line) and standard deviation (shaded area) for both the reconstruction error (in red) and training loss (in black) as the number of parameters $p$ increases, at different choices of input dimensions $d$ and number of dataset examples $n$. Statistics are computed across 10 distinct random seeds. Two distinct thresholds clearly emerge: $p \gg n$ for label fitting, and $p \gg dn$ for data reconstruction.

In Figure 2, the per-feature orthogonal residual (plotted in blue) remains large until the model crosses the threshold $p \approx dn$, after which it drops sharply (for $p \leq n$, the optimization converges to the non-degenerate case $P_{\hat{\Phi}} = I$, which makes the orthogonal residual trivially zero). Thus, this numerical evidence suggests that minimizing $\mathcal{L}(\hat{X})$ is sufficient to satisfy the hypotheses of our main theorems for $p \gg dn$, and hence to successfully reconstruct the training data. Equipped with these insights and a recipe for dataset reconstruction, we complement our theory with numerical results as discussed below.

# 5 NUMERICAL EXPERIMENTS

We initialize the rows $\hat{x}_i$ of $\hat{X}$ with i.i.d. standard Gaussian vectors. We then minimize $\mathcal{L}(\hat{X})$ in (11) with gradient descent with momentum, normalizing each row $\hat{x}_i$ after the update to constrain it on the $d$-dimensional sphere $\sqrt{d}\, \mathbb{S}^{d-1}$. In every experiment, we perform the optimization until the reconstruction loss converges to zero, *i.e.*, $\mathcal{L}(\hat{X}^*) = 0$ (up to machine precision). To quantify reconstruction quality, we report the average $\ell_2$ distance between the rows of the ground truth and reconstructed data matrices, modulo a permutation on the rows of the latter, *i.e.*,

$$\rho(X, \hat{X}^*) = \min_{\Pi \in \mathcal{P}_n} \frac{1}{n\sqrt{d}} \sum_{i=1}^{n} \left\| x_i - \hat{x}^*_{\Pi(i)} \right\|_2, \quad (12)$$

where $\mathcal{P}_n$ denotes the set of permutations of $[n]$. Obtaining $\Pi^*$ constitutes a classic linear assignment problem (Bertsekas, 1998), which we solve in polynomial time via the Hungarian method (Kuhn, 1955). We normalize our success metric with $n\sqrt{d}$ so that a reconstruction

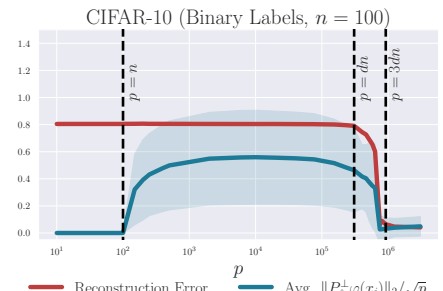

Figure 2: **Features of the training dataset $\Phi$ are spanned by the features of the reconstructed dataset $\hat{\Phi}$.** We consider the same setup as in Figure 1. For different values of $p$, we optimize until $\mathcal{L}(\hat{X}) = 0$, and report reconstruction error (in red, defined in Eq. (12)) and normalized residual $\|P_{\hat{\Phi}}^{\perp}\varphi(x_i)\|_2$ averaged over $i \in [n]$ (in blue), with their confidence interval at one standard deviation (shaded area). Further details and evidence are in Appendix E.1, see Figure 9.

can be considered successful as $\rho(X, \hat{X}^*)$ becomes much smaller than 1. As our main focus is on ReLU activations, whose odd Hermite coefficients (except $\mu_1$) are zero, recovered samples can appear with flipped sign (see Remark 1 and Figure 4). Accordingly, we also maximize $\rho(X, \hat{X}^*)$ over per-image sign flips. In the following, we present numerical results by first considering a synthetic setup (i.i.d. data uniformly distributed on the $d$-dimensional sphere), and then by reconstructing natural images from the CIFAR-10 dataset (Section 5.1). Finally, we explore the extent to which these phenomena are observed in neural networks trained with gradient descent (Section 5.2).

## 5.1 RANDOM FEATURES REGRESSION

**Synthetic data on the $d$-dimensional sphere.** We begin with a synthetic task, where the training data $X = [x_1, \ldots, x_n]^\top \in \mathbb{R}^{n \times d}$ are $n$ i.i.d. samples drawn uniformly from the sphere of radius $\sqrt{d}$. The labels $Y \in \mathbb{R}^n$ are given by $Y = Xg + \epsilon$, where $g \in \mathbb{R}^d$ has entries $g_i \sim \mathcal{N}(0, 1/d)$ and the

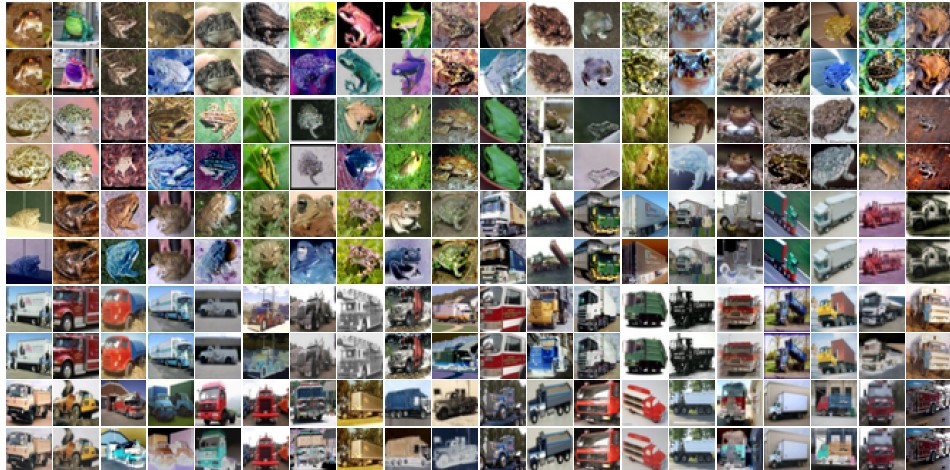

Figure 4: **Images reconstructed from an RF model with ReLU activation may have the wrong sign.** We repeat the experiment of Figure 1 using a different random seed and observe that reconstructions from ReLU models can appear as sign-flipped versions of original training data. This is due to the fact that ReLU has odd Hermite coefficients of order $\geq 3$ equal to zero.

noise $\epsilon \in \mathbb{R}^n$ is independent of the data $X$, with i.i.d. Gaussian entries with zero mean and variance 0.25. Figure 3 showcases the same trend for several choices of $d$ and $n$: at $p \geq n$, the training loss approaches zero, while the reconstruction error approaches zero when $p$ becomes larger than $dn$.

**Natural images with binary labels.**
The same phenomenon observed in the previous scenario carries over to natural images with binary labels $\{\pm 1\}$. For these experiments, we restrict the CIFAR-10 training split to the first 50 instances of the "*frog*" and "*truck*" classes, fitting several RF models with ReLU activation. In the left part of Figure 1, we can appreciate the same transitions at $p = n$ and $p \approx dn$ respectively for label fitting and data reconstruction; the right part of the figure then demonstrates that the re-

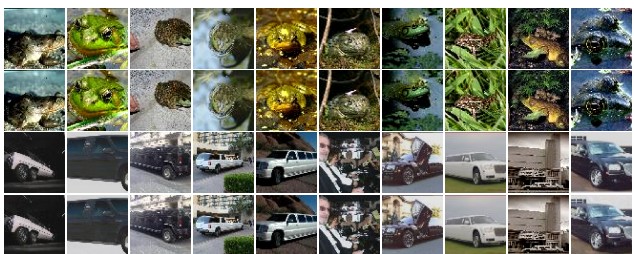

Figure 5: **Reconstruction of higher-dimensional images.** We repeat the same experiment of Figure 1 (right) using $n = 20$ samples from Tiny-ImageNet and a random features model with $p = 10dn$.

constructed dataset appears perceptually indistinguishable from the original training dataset. Notably, for ReLU the above visual match may fail due to a *sign error*. In fact, with a different seed, we reconstruct the negatives of some training images (Figure 4) even if the condition $p \gg dn$ is met and the loss in Eq. (11) converges. This aligns with Remark 1: ReLU violates Assumption 2 since $\mu_{2\ell+1} = 0$ for $\ell > 1$. In Figure 10 deferred to Appendix E.2, we show that the sign ambiguity disappears upon taking the activation $\phi(z) = \text{ReLU}(z) + \tanh(z)$, which has mixed-parity Hermite coefficients and, therefore, satisfies Assumption 2. In Figure 5, we display the reconstructed images from the same experiment on Tiny-ImageNet, which has input dimension $d = 3 \times 64 \times 64$, for $n = 20$ (additional details can be found in Figure 14 and Appendices D–E.3).

## 5.2 NEURAL NETWORKS TRAINED WITH GRADIENT DESCENT

Motivated by the RF baseline, we now turn our attention to finite-width networks, studying how *the number of parameters in the last layer* $p^{(L)}$ determines the ability to reconstruct the training dataset. Concretely, we train two-layer and deep residual networks with full-batch gradient descent, minimizing the square loss. Given that in this case the initialization is non-zero, we modify the reconstruction loss as $\mathcal{L}(\hat{X}) = \|P_{\hat{\Phi}}^{\perp}(\theta_*^{(L)} - \theta_0^{(L)})\|_2^2$, where $\theta_0^{(L)}$ and $\theta_*^{(L)}$ are respectively the parameters of the last layer at initialization and at the end of training. The feature matrix $\hat{\Phi}$ stacks the penultimate layer feature vectors $\varphi(\hat{x}_i)$. We assume access to trained weights of all layers $\{\theta_*^{(1)}, \dots, \theta_*^{(L)}\}$, the last layer at initialization $\theta_0^{(L)}$, and the neural network's computational graph.

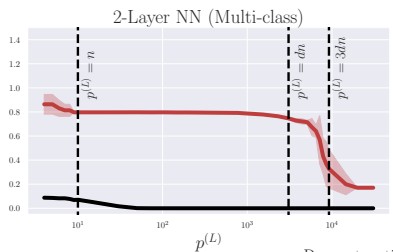
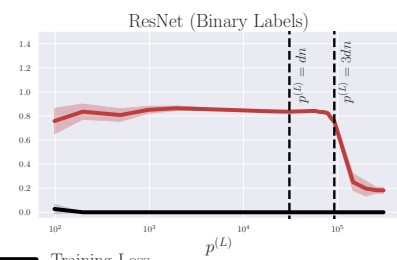

Figure 6: **Thresholds for label fitting and data reconstruction for neural networks trained with gradient descent on** $n = 10$ **CIFAR-10 images.** We consider regression with the square loss, training two-layer ReLU networks on the one-hot encoding of the 10-class labels (left) and ResNets on binary labels (right). We report mean (solid line) and standard deviation (shaded area) across 10 distinct random seeds for both reconstruction error (in red) and training loss (in black), as the number of parameters in the last layer $p^{(L)}$ increases.

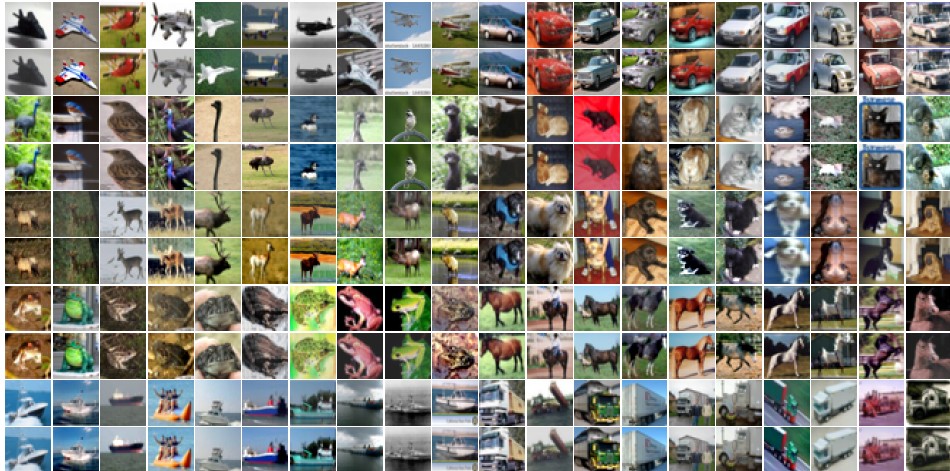

Figure 7: **Multi-class training data reconstructed from a neural network trained on CIFAR-10.** We train a two-layer ReLU network on $n = 100$ images from CIFAR-10 dataset (10 examples per class) with gradient descent. We cast the training procedure as multi-class regression on square loss, using one-hot encoded class labels as targets. The number of parameters in the last layer of the network is $p^{(L)} = 4dn$.

**Two-layer neural networks.** We consider a two-layer neural network $f_{\text{NN}}(x) = \theta^{(2)}\phi(\theta^{(1)}x)$, with $k$-class output and ReLU activation $\phi$. The weight matrices $\theta^{(1)} \in \mathbb{R}^{h \times d}$ and $\theta^{(2)} \in \mathbb{R}^{k \times h}$ have i.i.d. entries, initialized as $\theta^{(1)}_{i,j} \sim \mathcal{N}(0, 1/d)$, $\theta^{(2)}_{i,j} \sim \mathcal{N}(0, 1/h)$. In this setting, $p^{(L)} = k \times h$, where $h$ is the width of the network. In the left panel of Figure 6, we analyze the ability to reconstruct CIFAR-10 images with one-hot targets for each of the 10 classes. Although the output is no longer a scalar (as for random features), when $p^{(L)}$ becomes larger than $dn$, the reconstruction error starts decreasing. For all models reported in the plot, the total number of trainable parameters satisfies $p > n$ (even when $p^{(L)} < n$), so the model interpolates all training labels and the training loss is close to zero. As a confirmation of the quality of reconstructed images, we report in Figure 7 the reconstruction of $n = 100$ examples (10 per class), when $p^{(L)} = 4dn$. Also in this scenario, the reconstructed images are perceptually indistinguishable from original training data.

**Deep residual networks.** We conclude the experimental evaluation with residual architectures, probing how their structure (*i.e.*, residual connections and convolutions) affects reconstructability. We defer the formal definition of the model involved in these experiments to Appendix D.1. On CIFAR-10 (*frogs vs. trucks*), shown in the rightmost panel of Figure 6, ResNets display a transition for data reconstruction consistent with earlier experiments and happening after the $p^{(L)} = dn$ threshold. As for two-layer networks, the ResNets interpolate training labels even when $p^{(L)} < n$, since the total number of parameters $p$ is much larger than the number of samples $n$ for all models considered.

**Classification via logistic and cross-entropy loss.** We consider a synthetic task, where the training data $x_i$ are $n = 100$ i.i.d. samples drawn uniformly on the sphere of radius $\sqrt{d} = 10$. The labels are given by $y_i = \text{sign}(g^\top x_i)$, where $g \in \mathbb{R}^d$ has entries $g_i \sim \mathcal{N}(0, 1/d)$. We compute $\theta^*$ minimizing the logistic loss $\ell(\theta) := \sum_{i=1}^n \log(1 + e^{-y_i \varphi(x_i)^\top \theta})$ with gradient descent, and consider the same

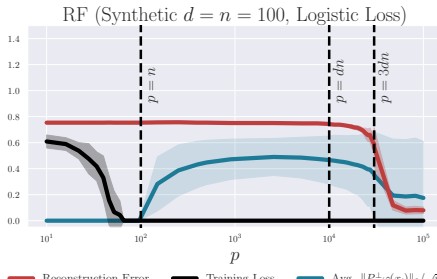 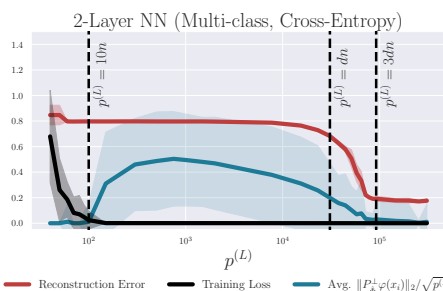

Figure 8: **Thresholds for data reconstruction on classification experiments.** (Left) We consider binary classification over $n = 100$ i.i.d. samples uniformly drawn from the $d$-dimensional sphere ($d = 100$). We train RF models with ReLU activation by minimizing the logistic loss with gradient descent. (Right) We consider multi-class classification and train two-layer ReLU networks on $n = 10$ CIFAR-10 images by minimizing cross-entropy loss with gradient descent. We report mean (solid line) and standard deviation (shaded area) for the reconstruction error (in red), for the training loss (in black) and for the per-feature orthogonal residual of projecting $\Phi$ onto $\text{span}\{\text{rows}(\hat{\Phi})\}$ (in blue). We indicate with $p^{(L)}$ the number of parameters in the last layer. Statistics are computed across 10 distinct random seeds.

reconstruction algorithm as in Eq. (11). In the left panel of Figure 8, we find the usual threshold $p \approx dn$ for successful reconstruction of the training set. In the right panel of Figure 8, we consider a two-layer neural network trained with cross-entropy loss on $n = 10$ samples from the 10 classes of CIFAR-10. and we see that the reconstruction algorithm yields similar results as the ones for regression in Figure 6. Further implementation details can be found in Appendix E.3, and the reconstructed images can be found in Figure 17.

**Additional ablation studies.** In Appendix E.3, we first show that the optimization of $\mathcal{L}(\hat{X})$ is robust to different choices of the learning rate $\eta$ (if it is sufficiently small). Then, we explore the setting where $\hat{X}$ has $\hat{n} \neq n$ rows, showing that the result of the optimization gives overlapping images in the case $\hat{n} < n$, and successfully reconstructs the full training set (plus some extra duplicates) in the case $\hat{n} > n$. We later show successful reconstruction from the parameters of a vision transformer, and give empirical evidence that adding a weight decay regularizer does not affect the law of reconstruction. Finally, we show that it is possible to reconstruct the data from a pruned neural network, albeit with higher values of $p$ depending on the sparsity.

## 6 CONCLUSIONS

This work studies data memorization, intended as the feasibility of reconstructing $n$ data samples of dimension $d$ from a trained model, focusing on the number of parameters $p$ at which this becomes possible. Our results on random features point to a *law of data reconstruction*, establishing the threshold $p \approx dn$. Remarkably, the reconstruction method grounded in our theoretical analysis is also successful in two-layer and deep residual networks. While our analysis assumes knowledge of the subspace of the training samples in feature space ($\text{span}\{\{\varphi(x_i)\}_{i=1}^n\}$), the experimental results demonstrate that, by optimizing the loss $\mathcal{L}(\hat{X})$ in Eq. (11), the whole training set is reconstructed when $p \gg dn$. This suggests two outstanding open problems to be tackled in future research: proving that *(i)* all the global optima of $\mathcal{L}(\hat{X})$ are permutations of the training dataset, and that *(ii)* the optimization problem can be efficiently solved with gradient methods despite the non-convexity of $\mathcal{L}(\hat{X})$. Another interesting direction is to understand what happens in the regime $n \ll p \ll dn$. On the one hand, Balle et al. (2022) indicate how to reconstruct *any single* training sample in this regime under certain model assumptions, given access to the final parameters and, additionally, to the remaining training data. On the other hand, we cautiously suspect that, without this additional knowledge, reconstructing the entire dataset is information-theoretically impossible when $p \ll dn$: at fixed machine precision, this would require recovering $\Theta(dn)$ bits from only $\Theta(p)$ bits, underscoring a hard limit on reconstructability. Importantly, this would not guarantee that *none* of the training samples is memorized. There is empirical evidence that learning models tend to memorize outliers (Feldman, 2020), and a possible direction for future work is to investigate if parts of the training data sampled from a heavy-tailed distribution (therefore not respecting Assumption 1) would result in memorization before the threshold $p \approx dn$.

## Ethics Statement

This paper presents work whose goal is to advance the field of machine learning. There are many potential societal consequences of our work, none which we feel must be specifically highlighted here.

## Reproducibility Statement

We have made significant efforts to ensure the reproducibility of our results. Full implementation details are provided in Appendix D, alongside practical design choices to address potential challenges in implementing our experiments. Additionally, we provide the complete codebase at `https://github.com/iurada/data-reconstruction-law`.

## Acknowledgements

M.M. is funded by the European Union (ERC, INF$^2$, project number 101161364). S.B. was supported by a Google PhD fellowship. L.I. acknowledges the grant received from the European Union Next-GenerationEU (Piano Nazionale di Ripresa E Resilienza (PNRR)) DM 351 on Trustworthy AI. T.T. & L.I. acknowledge the EU project ELSA - European Lighthouse on Secure and Safe AI. This study was carried out within the FAIR - Future Artificial Intelligence Research and received funding from the European Union Next-GenerationEU (PIANO NAZIONALE DI RIPRESA E RESILIENZA (PNRR) – MISSIONE 4 COMPONENTE 2, INVESTIMENTO 1.3 – D.D. 1555 11/10/2022, PE00000013). This manuscript reflects only the authors' views and opinions, neither the European Union nor the European Commission can be considered responsible for them. The authors would like to thank Yizhe Zhu for helpful discussions.

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

## A  ADDITIONAL NOTATIONS AND PRELIMINARIES

We use the definition of the Orlicz norm of order $\alpha$ of a real random variable $X$ as

$$\|X\|_{\psi_\alpha} := \inf\{t > 0 \; : \; \mathbb{E}\left[\exp(|X|^\alpha/t^\alpha)\right] \leq 2\}. \tag{13}$$

From this definition, it follows that $\left\||X|^\gamma\right\|_{\psi_{\alpha/\gamma}} = \|X\|_{\psi_\alpha}^\gamma$. We will say that a random variable $X$ is sub-Gaussian (sub-exponential) if its sub-Gaussian norm $\|X\|_{\psi_2}$ (sub-exponential norm $\|X\|_{\psi_1}$) is $O(1)$ (*i.e.* it does not increase with the scalings of the problem). Notice that if $X$ and $Y$ are scalar random variables, we have $\|XY\|_{\psi_1} \leq \|X\|_{\psi_2}\|Y\|_{\psi_2}$. We use the analogous definitions for vectors. In particular, let $X \in \mathbb{R}^n$ be a random vector, then $\|X\|_{\psi_2} := \sup_{\|u\|_2=1}\left\|u^\top X\right\|_{\psi_2}$ and $\|X\|_{\psi_1} := \sup_{\|u\|_2=1}\left\|u^\top X\right\|_{\psi_1}$. We note that if $X \in \mathbb{R}$ is sub-Gaussian (sub-exponential) and $\tau : \mathbb{R} \to \mathbb{R}$ is Lipschitz, we have that $\tau(X)$ is sub-Gaussian (sub-exponential) as well. Also, if a random variable is sub-Gaussian or sub-exponential, its $p$-th momentum is upper bounded by a constant (that might depend on $p$).

Given a matrix $A$, we indicate with $A_{i:}$ its $i$-th row, and with $A_{:j}$ its $j$-th column. Given a square matrix $A$, we denote by $\lambda_{\min}(A)$ its smallest eigenvalue. Given a matrix $A$ we indicate with $\|A\|_{\mathrm{op}}$ its operator (or spectral) norm, and with $\|A\|_F$ its Frobenius (or Hilbert-Schmidt) norm ($\|A\|_F^2 = \sum_{ij} A_{ij}^2$). Given two matrices $A, B \in \mathbb{R}^{m \times n}$, we denote by $A \circ B$ their Hadamard (component wise) product and by $A^{\circ l} = A \circ A^{\circ(l-1)}$ the $l$-th Hadamard power, with $A^{\circ 1} = A$.

## B  PROOF OF THEOREM 1

First, notice that, with overwhelming probability over $V$ and $X$, due to Lemma 4.5 in (Bombari & Mondelli, 2025a), we have

$$\lambda_{\min}\left(\Phi\Phi^\top\right) = \Omega(p). \tag{14}$$

In particular, notice that the lower bound on $n$ in their statement is not required to lower bound the smallest eigenvalue of the kernel. This implies that $\Phi\Phi^\top$ is full rank. Since for every $i \in [n]$ we have $\varphi(x_i) \in \mathrm{span}\{\mathrm{rows}(\hat{\Phi})\}$, we also have

$$\mathrm{span}\{\mathrm{rows}(\Phi)\} \subseteq \mathrm{span}\{\mathrm{rows}(\hat{\Phi})\}. \tag{15}$$

Thus, the equation above implies that a subspace of dimension $n$ ($\Phi\Phi^\top$ is full rank) is a subset of another subspace of dimension at most $n$. This implies that the two subspaces must be identical, which in turn implies, for every $i \in [n]$,

$$\varphi(\hat{x}_i) \in \mathrm{span}\{\mathrm{rows}(\Phi)\}. \tag{16}$$

Throughout the proof, we omit the subscript $\hat{i}$ to simplify the notation and we let $\hat{x}$ denote a row of $\hat{X}$ such that $\mathcal{L}(\hat{X}) = 0$. Then, due to (16), we will write

$$\varphi(\hat{x}) = \Phi^\top a = \sum_{i=1}^n \varphi(x_i)a_i, \tag{17}$$

where we introduced the vector $a \in \mathbb{R}^n$ containing the coefficients of $\varphi(\hat{x})$ written in the basis $\{\varphi(x_i)\}_{i=1}^n$.

Furthermore, in this section, we will introduce the shorthand $\tilde{\varphi}(x) = \tilde{\phi}(Vx)$, where $\tilde{\phi} : \mathbb{R} \to \mathbb{R}$ is a non-linearity that shares the same Hermite coefficients as $\phi$, but with the first $\tilde{\mu}_1 = 0$. This implies that $\tilde{\phi}(z) = \phi(z) - \mu_1 z$ is a Lipschitz function. We will use the shorthand $\tilde{\mu}^2 = \sum_{l=3}^\infty \mu_l^2$.

**Lemma B.1.** *For any $0 < t < p$ and any $i \in [n]$, we have that*

$$\left|\tilde{\varphi}(x_i)^\top \varphi(x_i) - \tilde{\mu}^2 p\right| = O\left(t\sqrt{p}\right), \tag{18}$$

*with probability at least $1 - 2\exp(-ct^2)$ over $V$. Furthermore, for any $i \neq j$, we have*

$$\left|\tilde{\varphi}(x_i)^\top \varphi(x_j)\right| = O\left(t\sqrt{p} + p\frac{\log^3 d}{d^{3/2}}\right), \tag{19}$$

*with probability at least $1 - 2\exp(-ct^2) - 2\exp(-c\log^2 d)$ over $V$ and $X$. Finally, we jointly have*

$$\sup_{\hat{x} \in \sqrt{d}\mathbb{S}^{d-1}} \left| \tilde{\varphi}(\hat{x})^\top \varphi(\hat{x}) - \tilde{\mu}^2 p \right| = O\left( \sqrt{pd}\log d + \frac{p}{d^2} \right), \tag{20}$$

$$\sup_{\hat{x} \in \sqrt{d}\mathbb{S}^{d-1}} \left| \tilde{\varphi}(x_i)^\top \varphi(\hat{x}) - \tilde{\varphi}(x_i)^\top \tilde{\varphi}(\hat{x}) \right| = O\left( \sqrt{pd}\log d + \frac{p}{d^2} \right), \tag{21}$$

$$\sup_{\hat{x} \in \sqrt{d}\mathbb{S}^{d-1}} \left| \varphi(x_i)^\top \tilde{\varphi}(\hat{x}) - \tilde{\varphi}(x_i)^\top \tilde{\varphi}(\hat{x}) \right| = O\left( \sqrt{pd}\log d + \frac{p}{d^2} \right), \tag{22}$$

*for all $i \in [n]$, with probability at least $1 - 2\exp(-cd\log^2 d)$ over $V$.*

*Proof.* For the first statement, we have

$$\begin{aligned}
\tilde{\varphi}(x_i)^\top \varphi(x_j) &= \sum_{k=1}^{p} \tilde{\phi}(v_k^\top x_i)\phi(v_k^\top x_j) \\
&= \sum_{k=1}^{p} \left( \tilde{\phi}(v_k^\top x_i)\phi(v_k^\top x_j) - \mathbb{E}_{v_k}\left[ \tilde{\phi}(v_k^\top x_i)\phi(v_k^\top x_j) \right] \right) \\
&\quad + p\, \mathbb{E}_v\left[ \tilde{\phi}(v^\top x_i)\phi(v^\top x_j) \right].
\end{aligned} \tag{23}$$

Let us analyze the two terms in the RHS separately. The first is the sum of $p$ independent, mean-0, sub-exponential random variables (in the probability space of $V$). This holds since both $\phi$ and $\tilde{\phi}$ are Lipschitz due to Assumption 2, and they arguments are sub-Gaussian. Then, due to Bernstein inequality (see Theorem 2.8.1 of (Vershynin, 2018)), for any $0 < t < p$, we have

$$\left| \sum_{k=1}^{p} \tilde{\phi}(v_k^\top x_i)\phi(v_k^\top x_j) - \mathbb{E}_{v_k}\left[ \tilde{\phi}(v_k^\top x_i)\phi(v_k^\top x_j) \right] \right| \leq t\sqrt{p}, \tag{24}$$

with probability at least $1 - 2\exp\left(-c_1 t^2\right)$ over $V$. For the second term in the RHS of (23), using the Hermite decomposition of $\phi$ and $\tilde{\phi}$, we have that

$$p\, \mathbb{E}_v\left[ \tilde{\phi}(v^\top x_i)\phi(v^\top x_j) \right] = p \sum_{l=3}^{+\infty} \mu_l^2 \frac{\left(x_i^\top x_j\right)^l}{d^l}. \tag{25}$$

Considering the case $i = j$, we readily obtain (18). For the case $i \neq j$, since the $x_i$-s are sampled independently from a sub-Gaussian distribution due to Assumption 1, and $\|x_i\|_2 = \sqrt{d}$ for every $i$, we have that

$$\max_{i \neq j} |x_i^\top x_j| \leq \sqrt{d}\log d, \tag{26}$$

with probability at least $1 - 2n^2 \exp(-c_2 \log^2 d) \geq 1 - 2\exp(-c_3 \log^2 d)$, due to Assumption 3. Then, with this probability, we have that

$$p \sum_{l=3}^{+\infty} \mu_l^2 \frac{\left(x_i^\top x_j\right)^l}{d^l} \leq p\frac{\left(x_i^\top x_j\right)^3}{d^3} \sum_{l=3}^{+\infty} \mu_l^2 \leq \tilde{\mu}^2 p\frac{\log^3 d}{d^{3/2}}, \tag{27}$$

for every $i \neq j$, which gives (19).

Let us now consider an $\epsilon\sqrt{d}$-net of $\sqrt{d}\mathbb{S}^{d-1}$, namely $\{x_m^\epsilon\}_{m=1}^{M}$, such that for any $x \in \sqrt{d}\mathbb{S}^{d-1}$ there exists $m \in [M]$ such that $\|x - x_m^\epsilon\|_2 \leq \epsilon\sqrt{d}$. Due to Corollary 4.2.13 in (Vershynin, 2018), for $\epsilon < 1$ we have that the net can be chosen such that $M \leq (3/\epsilon)^d$. Notice that, for any $m \in [M]$, the same argument leading to (18) yields

$$\left| \tilde{\varphi}(x_m^\epsilon)^\top \varphi(x_m^\epsilon) - \tilde{\mu}^2 p \right| = O(\sqrt{pd}\log d), \tag{28}$$

with probability at least $1 - 2\exp(-c_3 d\log^2 d)$, after setting $t = \sqrt{d}\log d$. Setting $\epsilon = 1/d^2$, and performing a union bound on all $m \in [M]$, we have that the previous statement holds uniformly with probability at least

$$1 - 2(3/\epsilon)^d \exp(-c_3 d\log^2 d) = 1 - 2\exp((\log(3/\epsilon) - c_3 \log^2 d)d) \geq 1 - 2\exp(-c_4 d\log^2 d), \tag{29}$$

where $c_4$ is an absolute constant. By Lemma C.3 in (Bombari & Mondelli, 2024) we also have that for any $m \in [M]$, we jointly have

$$\|\varphi(x_m^\epsilon)\|_2 = O(\sqrt{p}), \qquad \|\tilde{\varphi}(x_m^\epsilon)\|_2 = O(\sqrt{p}), \tag{30}$$

with probability at least $1 - 2\exp(-c_5 p)$. Taking a union bound for all $m \in [M]$, (30) holds uniformly over the net with probability at least $1 - 2\exp(-c_6 p)$ due to Assumption 3. Since $\epsilon = 1/d^2$, for any $\hat{x}$, there exists $m$ such that $\|\hat{x} - x_m^\epsilon\|_2 \le 1/d^{3/2}$. This implies

$$\sup_{\hat{x} \in \sqrt{d}\,\mathbb{S}^{d-1}} \|\varphi(\hat{x})\|_2 \le \max_{m \in [M]} \|\varphi(x_m^\epsilon)\|_2 + \sup_{\hat{x} \in \sqrt{d}\,\mathbb{S}^{d-1}} \|\varphi(\hat{x}) - \varphi(x_m^\epsilon)\|_2$$

$$\le \max_{m \in [M]} \|\varphi(x_m^\epsilon)\|_2 + L\,\|V\|_{\mathrm{op}} \sup_{\hat{x} \in \sqrt{d}\,\mathbb{S}^{d-1}} \|\hat{x} - x_m^\epsilon\|_2 \tag{31}$$

$$= O\left(\sqrt{p} + L\sqrt{\frac{p}{d}}\frac{1}{d^{3/2}}\right) = O(\sqrt{p}),$$

where the third step follows from $\|V\|_{\mathrm{op}} = O(\sqrt{p/d})$, which holds with probability at least $1 - 2\exp(-c_7 p)$ due to Theorem 4.4.5 in (Vershynin, 2018). Then, we have

$$\sup_{\hat{x} \in \sqrt{d}\,\mathbb{S}^{d-1}} \left|\tilde{\varphi}(x_m^\epsilon)^\top \varphi(x_m^\epsilon) - \tilde{\varphi}(\hat{x})^\top \varphi(\hat{x})\right| \le \sup_{\hat{x} \in \sqrt{d}\,\mathbb{S}^{d-1}} \left|\tilde{\varphi}(x_m^\epsilon)^\top \varphi(x_m^\epsilon) - \tilde{\varphi}(x_m^\epsilon)^\top \varphi(\hat{x})\right|$$

$$+ \sup_{\hat{x} \in \sqrt{d}\,\mathbb{S}^{d-1}} \left|\tilde{\varphi}(x_m^\epsilon)^\top \varphi(\hat{x}) - \tilde{\varphi}(\hat{x})^\top \varphi(\hat{x})\right|$$

$$\le \max_{m \in [M]} \|\tilde{\varphi}(x_m^\epsilon)\|_2\, L\, \|V\|_{\mathrm{op}} \sup_{\hat{x} \in \sqrt{d}\,\mathbb{S}^{d-1}} \|x_m^\epsilon - \hat{x}\|_2$$

$$+ \sup_{\hat{x} \in \sqrt{d}\,\mathbb{S}^{d-1}} \|\varphi(\hat{x})\|_2\, \tilde{L}\, \|V\|_{\mathrm{op}} \sup_{\hat{x} \in \sqrt{d}\,\mathbb{S}^{d-1}} \|x_m^\epsilon - \hat{x}\|_2$$

$$= O\left(\sqrt{p}\frac{\sqrt{p}}{\sqrt{d}}\frac{1}{d^{3/2}}\right) = O\left(\frac{p}{d^2}\right). \tag{32}$$

Thus, merging (28), (29) and (32) yields

$$\sup_{\hat{x} \in \sqrt{d}\,\mathbb{S}^{d-1}} \left|\tilde{\varphi}(\hat{x})^\top \varphi(\hat{x}) - \tilde{\mu}^2 p\right| = O\left(\sqrt{pd}\log d + \frac{p}{d^2}\right), \tag{33}$$

with probability at least $1 - 2\exp(-c_8 d\log^2 d)$ over $V$, which proves (20).

Let us now consider a fixed $i \in [n]$ and $m \in [M]$. We have

$$\tilde{\varphi}(x_i)^\top \varphi(x_m^\epsilon) - \tilde{\varphi}(x_i)^\top \tilde{\varphi}(x_m^\epsilon) = \sum_{k=1}^{p} \left(\tilde{\phi}(v_k^\top x_i)\phi(v_k^\top x_m^\epsilon) - \mathbb{E}_{v_k}\left[\tilde{\phi}(v_k^\top x_i)\phi(v_k^\top x_m^\epsilon)\right]\right)$$

$$- \sum_{k=1}^{p} \left(\tilde{\phi}(v_k^\top x_i)\tilde{\phi}(v_k^\top x_m^\epsilon) - \mathbb{E}_{v_k}\left[\tilde{\phi}(v_k^\top x_i)\tilde{\phi}(v_k^\top x_m^\epsilon)\right]\right)$$

$$+ p\,\mathbb{E}_v\left[\tilde{\phi}(v^\top x_i)\phi(v^\top x_m^\epsilon)\right] - p\,\mathbb{E}_v\left[\tilde{\phi}(v^\top x_i)\tilde{\phi}(v^\top x_m^\epsilon)\right]. \tag{34}$$

The last line is equal to 0, due to the Hermite coefficients of $\phi$ and $\tilde{\phi}$. The first two lines of the RHS can be bounded separately via Bernstein inequality as in (24), giving

$$\left|\tilde{\varphi}(x_i)^\top \varphi(x_m^\epsilon) - \tilde{\varphi}(x_i)^\top \tilde{\varphi}(x_m^\epsilon)\right| = O(\sqrt{p}\,t), \tag{35}$$

with probability at least $1 - 2\exp(-c_9 t^2)$ over $V$, for any $0 < t < p$. As done in (29), considering again $\epsilon = 1/d^2$, we have that the previous bound holds uniformly for every $i \in [n]$ and for every $m \in [M]$, after setting $t = \sqrt{d}\log d$, with probability at least $1 - 2\exp(-c_{10} d\log^2 d)$. Then, a similar argument as the one used in (32) yields

$$\sup_{\hat{x} \in \sqrt{d}\,\mathbb{S}^{d-1}} \left|\tilde{\varphi}(x_i)^\top \varphi(\hat{x}) - \tilde{\varphi}(x_i)^\top \tilde{\varphi}(\hat{x})\right| = O\left(\sqrt{pd}\log d + \frac{p}{d^2}\right), \tag{36}$$

with probability at least $1 - 2\exp(-c_{11} d\log^2 d)$. This proves (21). The proof of (22) is analogous and the argument is complete. $\qquad\square$

**Lemma B.2.** *Let $\hat{x}$ be a generic row of $\hat{X}$ such that $\mathcal{L}(\hat{X}) = 0$ and $a \in \mathbb{R}^n$ be defined according to (17). Then, we have that, for any $i \in [n]$,*

$$\left| \tilde{\varphi}(x_i)^\top \varphi(\hat{x}) - \tilde{\mu}^2 p a_i \right| = O\left( (\|a\|_2 + 1)\left( \sqrt{pd}\log d + \sqrt{pn}\log d + p\frac{\sqrt{n}\log^3 d}{d^{3/2}} \right) \right), \quad (37)$$

*with probability at least $1 - 2\exp(-c\log^2 d)$ over $V$ and $X$.*

*Proof.* For any $i \in [n]$, (17) implies

$$\begin{aligned}
\tilde{\varphi}(x_i)^\top \varphi(\hat{x}) = \tilde{\varphi}(x_i)^\top \Phi^\top a = \sum_{j=1}^n \tilde{\varphi}(x_i)^\top \varphi(x_j) a_j \\
= \tilde{\mu}^2 p a_i + \left( \tilde{\varphi}(x_i)^\top \varphi(x_i) a_i - \tilde{\mu}^2 p a_i \right) + \sum_{j \neq i} \tilde{\varphi}(x_i)^\top \varphi(x_j) a_j.
\end{aligned} \quad (38)$$

This implies

$$\begin{aligned}
\left| \tilde{\varphi}(x_i)^\top \varphi(\hat{x}) - \tilde{\mu}^2 p a_i \right| &\leq \left| \tilde{\varphi}(x_i)^\top \varphi(x_i) - \tilde{\mu}^2 p \right| |a_i| + \sqrt{\sum_{j \neq i} (\tilde{\varphi}(x_i)^\top \varphi(x_j))^2} \|a\|_2 \\
&= O\left( \|a\|_2 \left( \sqrt{p}\log d + \sqrt{n\left( p\log^2 d + p^2\frac{\log^6 d}{d^3} \right)} \right) \right) \quad (39) \\
&= O\left( \|a\|_2 \left( \sqrt{p}\log d + \sqrt{pn}\log d + p\frac{\sqrt{n}\log^3 d}{d^{3/2}} \right) \right),
\end{aligned}$$

with probability at least $1 - 2\exp\left(-c_1\log^2 d\right)$ over $V$ and $X$, where the second step holds due to the first two equations in the statement of Lemma B.1. Then, due to the fourth equation in the statement of Lemma B.1, we have

$$\left| \tilde{\varphi}(x_i)^\top \varphi(\hat{x}) - \tilde{\varphi}(x_i)^\top \tilde{\varphi}(\hat{x}) \right| = O\left( \sqrt{pd}\log d + \frac{p}{d^2} \right), \quad (40)$$

which, together with (39), gives the desired result. $\square$

**Lemma B.3.** *We have that*

$$\left| \|a\|_2^2 - 1 \right| = O\left( \sqrt{\frac{dn}{p}}\log d + \frac{\log^3 d}{\sqrt{d}} \right), \quad (41)$$

*with probability at least $1 - 2\exp(-c\log^2 d)$.*

*Proof.* (17) directly implies

$$\tilde{\varphi}(\hat{x})^\top \varphi(\hat{x}) = \tilde{\varphi}(\hat{x})^\top \Phi^\top a_i = \sum_{j=1}^n \tilde{\varphi}(\hat{x})^\top \varphi(x_j) a_j. \quad (42)$$

Due to the third and fifth equations in the statement of Lemma B.1, we respectively have that

$$\left| \tilde{\varphi}(\hat{x})^\top \varphi(\hat{x}) - \tilde{\mu}^2 p \right| = O\left( \sqrt{pd}\log d + \frac{p}{d^2} \right), \quad (43)$$

$$\left| \varphi(x_i)^\top \tilde{\varphi}(\hat{x}) - \tilde{\varphi}(x_i)^\top \tilde{\varphi}(\hat{x}) \right| = O\left( \sqrt{pd}\log d + \frac{p}{d^2} \right), \quad (44)$$

with probability at least $1 - 2\exp(-c_1 d\log^2 d)$. Then, by Cauchy-Schwartz inequality, (44) yields

$$\left| \sum_{i=1}^n \tilde{\varphi}(\hat{x})^\top \varphi(x_i) a_i - \sum_{i=1}^n \tilde{\varphi}(\hat{x})^\top \tilde{\varphi}(x_i) a_i \right| = O\left( \|a\|_2 \left( \sqrt{pdn}\log d + \frac{p\sqrt{n}}{d^2} \right) \right). \quad (45)$$

Again by Cauchy-Schwartz inequality, we have

$$\left| \sum_{i=1}^{n} a_i \left( \tilde{\varphi}(\hat{x})^\top \tilde{\varphi}(x_i) - \tilde{\mu}^2 p a_i \right) \right| =$$

$$= O \left( \|a\|_2 \left( \|a\|_2 + 1 \right) \left( \sqrt{pnd} \log d + \sqrt{pn} \log d + pn \frac{\log^3 d}{d^{3/2}} \right) \right) \tag{46}$$

$$= O \left( \|a\|_2 \left( \|a\|_2 + 1 \right) \left( \sqrt{pnd} \log d + p \frac{\log^3 d}{\sqrt{d}} \right) \right),$$

with probability at least $1 - 2 \exp(-c_2 \log^2 d)$, due to Lemma B.2, where the last step is a consequence of Assumption 3. Then, merging (42), (43), (45) and (46), an application of the triangle inequality yields

$$\tilde{\mu}^2 p \left| \|a\|_2^2 - 1 \right| = O \left( \left( \|a\|_2^2 + 1 \right) \left( \sqrt{pdn} \log d + p \frac{\log^3 d}{\sqrt{d}} \right) \right). \tag{47}$$

Notice that, if $\|a\|_2 = \omega(1)$, the LHS of the previous equation would be $\Omega(p \|a\|_2^2)$, while the RHS would be $o(p \|a\|_2^2)$ due to Assumption 3. Then, we necessarily have that $\|a\|_2 = O(1)$, which yields

$$\left| \|a\|_2^2 - 1 \right| = O \left( \sqrt{\frac{dn}{p}} \log d + \frac{\log^3 d}{\sqrt{d}} \right), \tag{48}$$

with probability at least $1 - 2 \exp(-c_3 \log^2 d)$, which gives the desired result. $\qquad\square$

**Lemma B.4.** *Let $\hat{x} \in \sqrt{d} \, \mathbb{S}^{d-1}$ a generic vector, and let $0 \le C \le 1$ be defined as*

$$C = \max_i \left| \frac{x_i^\top \hat{x}}{d} \right|. \tag{49}$$

*Then, with probability at least $1 - 2 \exp \left( -c\sqrt{d} \right)$ over $X$, we have that*

$$\left\| \frac{(X\hat{x})^{\circ l}}{d^l} \right\|_2 \le C^{l-1} + O \left( d^{-0.1} \right), \tag{50}$$

*uniformly for every $l \ge 2$.*

*Proof.* Fix $0 \le \delta \le 1$, and consider the values of $i \in [n]$ such that

$$\left| \frac{x_i^\top \hat{x}}{d} \right| > \delta. \tag{51}$$

Define $M \subseteq [n]$ as the set containing all the values of $i$ that satisfy the inequality above, $m = |M|$ as the cardinality of $M$, and $X_\delta \in \mathbb{R}^{m \times d}$ as the matrix that contains in its rows all and only the $x_i$-s such that $i \in M$. Note that $X^\top$ is a matrix with independent sub-Gaussian columns, with fixed $\ell_2$ norm equal to $\sqrt{d}$. Then, Theorem 1.3 in (Plan & Vershynin, 2025) yields

$$\left| \|X\|_{\mathrm{op}} - \sqrt{d} \right| = O(\sqrt{n}), \tag{52}$$

with probability at least $1 - 2 \exp(-c_1 n)$. This, due to Assumption 3, guarantees that $\left\| X/\sqrt{d} \right\|_{\mathrm{op}} = O(1)$, and therefore

$$\delta^2 m \le \left\| \frac{X\hat{x}}{d} \right\|_2^2 \le \left\| \frac{X}{\sqrt{d}} \right\|_{\mathrm{op}}^2 \left\| \frac{\hat{x}}{\sqrt{d}} \right\|_2^2 = O(1), \tag{53}$$

which implies

$$m = O \left( \frac{1}{\delta^2} \right). \tag{54}$$

Consider all the possible subsets of size $m$ of $[n]$. There are in total

$$\binom{n}{m} \leq n^m \leq \exp(m \log n) \leq \exp\left(C_1 \frac{\log d}{\delta^2}\right) \tag{55}$$

such subsets, where $C$ is an absolute constant, and where we used Assumption 3. For each of these subsets, the operator norm of the matrix $X_s \in \mathbb{R}^{m \times d}$ with rows indexed by $M$ is

$$\left| \|X_s\|_{\mathrm{op}} - \sqrt{d} \right| \leq C_2 \left(\sqrt{m} + t\right), \tag{56}$$

with probability $1 - 2\exp(-c_2 t^2)$, again due to Theorem 1.3 in (Plan & Vershynin, 2025). Then, performing a union bound over all subsets and setting $t = \sqrt{2C_1/c_2} \frac{\sqrt{\log d}}{\delta}$, we have that all matrices with rows belonging to a generic subset of size $m$ of the rows of $X$ respect (56) with probability at least $1 - 2\exp\left(-C_1 \frac{\log d}{\delta^2}\right)$. In particular, with this probability, we also have

$$\left| \|X_\delta\|_{\mathrm{op}} - \sqrt{d} \right| \leq C_2 \left(\sqrt{m} + t\right) = O\left(\frac{\sqrt{\log d}}{\delta}\right), \tag{57}$$

where in the second step we used (54). Then, we have that

$$\left\| \frac{(X\hat{x})^{\circ l}}{d^l} \right\|_2^2 \leq \left\| \frac{(X_\delta \hat{x})^{\circ l}}{d^l} \right\|_2^2 + (n - m)\delta^{2l} \leq C^{2l-2} \left\| \frac{X_\delta \hat{x}}{d} \right\|_2^2 + n\delta^4, \tag{58}$$

where in the last step we used that $l \geq 2$, and the definition of $C$ in (49). Setting $\delta = d^{-0.3}$, Assumption 3 gives $n\delta^4 = O(d^{-0.2})$, which yields

$$\left\| \frac{X_\delta \hat{x}}{d} \right\|_2 \leq \left\| X_\delta/\sqrt{d} \right\|_{\mathrm{op}} \leq 1 + \left\| \|X_\delta/\sqrt{d}\|_{\mathrm{op}} - 1 \right\| \leq 1 + C_3 \frac{\sqrt{\log d} + 1}{\sqrt{d} \, d^{-0.3}}, \tag{59}$$

where the last step holds due to (57) with probability at least $1 - 2\exp\left(-C_1 \frac{\log d}{d^{-0.6}}\right) \geq 1 - 2\exp\left(-c_3\sqrt{d}\right)$. Thus, plugging in (58), the thesis readily follows. $\square$

**Proof of Theorem 1.** As done in Lemma B.1, consider the $\sqrt{d}\,\epsilon$-net of the sphere $\sqrt{d}\,\mathbb{S}^{d-1}$, with $\epsilon = 1/d$. For any element $x_m^\epsilon$ of this set (with a fixed index $m \in [M]$, where $M$ denotes the cardinality of the net), (17) yields

$$\tilde{\varphi}(x_m^\epsilon)^\top \varphi(\hat{x}) = \tilde{\varphi}(x_m^\epsilon)^\top \Phi^\top a = \sum_{i=1}^n \tilde{\varphi}(x_m^\epsilon)^\top \varphi(x_i) a_i, \tag{60}$$

where each term of the sum above reads

$$\tilde{\varphi}(x_m^\epsilon)^\top \varphi(x_i) = \sum_{k=1}^p \left( \tilde{\phi}(v_k^\top x_m^\epsilon)\phi(v_k^\top x_i) - \mathbb{E}_{v_k}\left[ \tilde{\phi}(v_k^\top x_m^\epsilon)\phi(v_k^\top x_i) \right] \right) + p\,\mathbb{E}_v\left[ \tilde{\phi}(v^\top x_m^\epsilon)^\top \phi(v^\top x_i) \right]. \tag{61}$$

By Bernstein inequality (see the same argument as in (24)), we have that

$$\sum_{k=1}^p \left( \tilde{\phi}(v_k^\top x_m^\epsilon)\phi(v_k^\top x_i) - \mathbb{E}_{v_k}\left[ \tilde{\phi}(v_k^\top x_m^\epsilon)\phi(v_k^\top x_i) \right] \right) = O\left(\sqrt{pd}\log d\right), \tag{62}$$

with probability at least $1 - 2\exp\left(-c_1 d \log^2 d\right)$. Performing a union bound over the elements of the net, we have that (62) holds uniformly for all $i \in [n]$ and all $m \in [M]$ with probability at least $1 - 2\exp\left(-c_2 d \log^2 d\right)$ (see the argument prior to (29)). Furthermore, the Hermite decomposition of $\phi$ and $\tilde{\phi}$ gives

$$\mathbb{E}_v\left[ \tilde{\phi}(v^\top x_m^\epsilon)\phi(v^\top x_i) \right] = \sum_{l=3}^{+\infty} \mu_l^2 \frac{(x_i^\top x_m^\epsilon)^l}{d^l}, \tag{63}$$

which thus allows us to write

$$\left| \sum_{i=1}^{n} \tilde{\varphi}(x_m^\epsilon)^\top \varphi(x_i) a_i - p \sum_{i=1}^{n} a_i \sum_{l=3}^{+\infty} \mu_l^2 \frac{(x_i^\top x_m^\epsilon)^l}{d^l} \right| = O\left( \|a\|_2 \sqrt{pdn} \log d \right)$$
$$= O\left( \sqrt{pdn} \log d \right), \tag{64}$$

where the first step follows from (61), (62) and (63), and an application of Cauchy Schwartz inequality; the second step holds due to Lemma B.3 with probability at least $1 - 2\exp\left(-c_3 \log^2 d\right)$.

By the definition of the net, there exists $m \in [M]$ such that $\|x_m^\epsilon - \hat{x}\|_2 \leq 1/\sqrt{d}$. Fixing such $m$, we have

$$\left| \tilde{\mu}^2 p - p \sum_{i=1}^{n} a_i \sum_{l=3}^{+\infty} \mu_l^2 \frac{(x_i^\top x_m^\epsilon)^l}{d^l} \right| \leq \left| \tilde{\mu}^2 p - \tilde{\varphi}(\hat{x})^\top \varphi(\hat{x}) \right| + \left| \tilde{\varphi}(\hat{x})^\top \varphi(\hat{x}) - \tilde{\varphi}(x_m^\epsilon)^\top \varphi(\hat{x}) \right|$$

$$+ \left| \sum_{i=1}^{n} \tilde{\varphi}(x_m^\epsilon)^\top \varphi(x_i) a_i - p \sum_{i=1}^{n} a_i \sum_{l=3}^{+\infty} \mu_l^2 \frac{(x_i^\top x_m^\epsilon)^l}{d^l} \right| \tag{65}$$

$$= O\left( \sqrt{pd} \log d + \frac{p}{d} + \frac{p}{d} + \sqrt{pdn} \log d \right)$$

$$= O\left( \sqrt{pdn} \log d + \frac{p}{d} \right),$$

due to the third equation in the statement of Lemma B.1, an argument equivalent to the one in (32), and (64). Considering the union bound on these high probability events, we have that (65) holds with probability at least $1 - 2\exp\left(-c_4 \log^2 d\right)$.

Let's suppose we have

$$\max_i \left| \frac{\hat{x}^\top x_i}{d} \right| \leq C < 1, \tag{66}$$

where $C$ is an absolute constant (independent of $d, n, p$). Thus,

$$\max_i \left| \frac{x_m^\epsilon {}^\top x_i}{d} \right| \leq \max_i \left| \frac{\hat{x}^\top x_i}{d} \right| + \max_i \left| \frac{\|x_m^\epsilon - \hat{x}\|_2 \|x_i\|_2}{d} \right| \leq C + \frac{1}{d} \leq C_\epsilon < 1, \tag{67}$$

where the last inequality holds for sufficiently large $d$. Then, we have

$$\left| \sum_{i=1}^{n} a_i \sum_{l=3}^{+\infty} \mu_l^2 \frac{(x_i^\top x_m^\epsilon)^l}{d^l} \right| \leq \|a\|_2 \sum_{l=3}^{+\infty} \mu_l^2 \left\| \frac{(X x_m^\epsilon)^{\circ l}}{d^l} \right\|_2$$

$$\leq \left( 1 + C_1 \left( \sqrt{\frac{dn}{p}} \log d + \frac{\log^3 d}{\sqrt{d}} \right) \right) \sum_{l=3}^{+\infty} \mu_l^2 \left( C_\epsilon^{l-1} + C_2 d^{-0.1} \right) \tag{68}$$

$$\leq \tilde{\mu}^2 C_\epsilon^2 + C_3 \left( \sqrt{\frac{dn}{p}} \log d + d^{-0.1} \right),$$

where we applied Cauchy Schwartz inequality separately for every $l$ in the first step and used Lemmas B.3 and B.4 in the second step. Here, $C_1$ and $C_2$ denote two positive absolute constants, and the inequality holds with probability at least $1 - 2\exp\left(-c_5 \log^2 d\right)$. Plugging (68) in (65) yields

$$\tilde{\mu}^2 \left( 1 - C_\epsilon^2 \right) = O\left( \sqrt{\frac{dn}{p}} \log d + d^{-0.1} \right) = o(1), \tag{69}$$

where the last step is a consequence of Assumption 3, which provides a contradiction with the hypothesis in (66), implying that

$$\left| 1 - \max_i \left| \frac{\hat{x}^\top x_i}{d} \right| \right| = o(1), \tag{70}$$

with probability at least $1 - 2\exp\left(-c_5 \log^2 d\right)$.

Let $j = \arg\max_i \left| \hat{x}^\top x_i \right|$ (taking the smallest index if there are multiple indices that maximize this value), and suppose $\hat{x}^\top x_i \geq 0$. Then, the law of cosines yields

$$\|\hat{x} - x_j\|_2 = \sqrt{2d\left(1 - \frac{\hat{x}^\top x_j}{d}\right)} = o\left(\sqrt{d}\right), \tag{71}$$

where the last step holds due to (70). Thus, for $k \neq j$ we have

$$\left| \frac{x_k^\top \hat{x}}{d} \right| = \left| \frac{x_k^\top x_j}{d} + \frac{x_k^\top (\hat{x} - x_j)}{d} \right| \leq \frac{\left| x_k^\top x_j \right|}{d} + \frac{\|x_k\|_2 \|\hat{x} - x_j\|_2}{d} = o(1), \tag{72}$$

where the last step holds with probability at least $1 - 2\exp\left(-c_6 \log^2 d\right)$ due to (71) and (26). In the case $\hat{x}^\top x_i < 0$ the same argument with $-x_j$ instead of $x_j$ yields the same thesis. Thus, comparing with (70), we have that $\arg\max_i \left| \hat{x}^\top x_i \right|$ has a unique solution $j$, and all other indices are such that $\left| \hat{x}^\top x_i \right| / d = o(1)$.

This proves that $\hat{x}$ is aligned with $x_i$. It remains to prove that $\hat{x}$ also has the correct sign (i.e., it is close to $x_i$ and not to $-x_i$). To do so, following the same approach that led to (68), we have that

$$\left| \sum_{i \neq j} a_i \sum_{l=3}^{+\infty} \mu_l^2 \frac{\left(x_i^\top x_m^\epsilon\right)^l}{d^l} \right| = o(1), \tag{73}$$

with probability at least $1 - 2\exp\left(-c_7 \log^2 d\right)$. Thus, comparing with (65), we get

$$\left| \tilde{\mu}^2 - a_j \sum_{l=3}^{+\infty} \mu_l^2 \frac{\left(x_j^\top x_m^\epsilon\right)^l}{d^l} \right| = o(1). \tag{74}$$

Since $|a_j| \leq 1 + o(1)$ by Lemma B.3, (74) can hold only if $\left(x_j^\top x_m^\epsilon\right)^l$ have all the same sign for all $l \geq 3$ such that $\mu_l \neq 0$. By Assumption 2, this is possible only if $x_j^\top x_m^\epsilon > 0$, which concludes the argument. $\qquad\square$

## C  PROOF OF THEOREM 2

Due to Theorem 1, we have that, with overwhelming probability, every $\hat{x}_{\hat{\imath}}$ has a unique "closest" row vector in $X$, such that

$$\left| 1 - \frac{\hat{x}_{\hat{\imath}}^\top x_i}{d} \right| = o(1). \tag{75}$$

Then, if we consider the case $n = 2$, either both samples are reconstructed, or the same training sample is reconstructed twice. By contradiction, let us suppose the latter hypothesis, which without loss of generality can be framed as the first sample $x_1$ being reconstructed twice.

Since we have that $\varphi(x_2) \in \text{span}\{\text{rows}(\hat{\Phi})\}$, which means that there exist two real numbers $a_1$ and $a_2$ such that

$$\varphi(x_2) = a_1 \varphi(x_1 + \epsilon_1) + a_2 \varphi(x_1 + \epsilon_2), \tag{76}$$

where

$$\|x_1 + \epsilon_1\|_2 = \|x_1 + \epsilon_2\|_2 = \sqrt{d}, \tag{77}$$

as we are considering data reconstruction on the sphere. From now on, we will always assume that (76) and (77) hold. We will often consider the following expansion:

$$\begin{aligned}
\varphi(x_2) &= a_1 \varphi(x_1 + \epsilon_1) + a_2 \varphi(x_1 + \epsilon_2) \\
&= a_1 \varphi(x_1 + \epsilon_1) + a_2 \varphi(x_1 + \epsilon_1 + (\epsilon_2 - \epsilon_1)) \\
&= (a_1 + a_2)\varphi(x_1 + \epsilon_1) \\
&\quad + a_2 \left((\varphi(x_1 + \epsilon_2) - \varphi(x_1 + \epsilon_1)) - \phi'(V(x_1 + \epsilon_1)) \circ (V(\epsilon_2 - \epsilon_1))\right) \\
&\quad + a_2 \, \phi'(V(x_1 + \epsilon_1)) \circ (V(\epsilon_2 - \epsilon_1)),
\end{aligned} \tag{78}$$

which can be used since $\phi$ admits first derivative according to Assumption 2. Note that

$$0 = \|x_1 + \epsilon_1\|_2^2 - \|x_1\|_2^2 = 2x_1^\top \epsilon_1 + \|\epsilon_1\|_2^2, \tag{79}$$

which yields

$$2\left|x_1^\top \epsilon_1\right| = \|\epsilon_1\|_2^2, \tag{80}$$

with the same relation holding for $\epsilon_2$. Similarly, we have

$$0 = \|x_1 + \epsilon_1\|_2^2 - \|x_1 + \epsilon_2\|_2^2 = 2x_1^\top (\epsilon_1 - \epsilon_2) + (\|\epsilon_1\|_2 + \|\epsilon_2\|_2)(\|\epsilon_1\|_2 - \|\epsilon_2\|_2), \tag{81}$$

which yields

$$2\left|x_1^\top (\epsilon_1 - \epsilon_2)\right| \le (\|\epsilon_1\|_2 + \|\epsilon_2\|_2)\|\epsilon_1 - \epsilon_2\|_2. \tag{82}$$

We will use the following notation

$$\epsilon = \frac{\max\{\|\epsilon_1\|_2, \|\epsilon_2\|_2\}}{\sqrt{d}} = O(1), \tag{83}$$

and

$$\delta = \frac{\|\epsilon_2 - \epsilon_1\|_2}{\sqrt{d}} = O(1). \tag{84}$$

Notice that, by definition, we have $\delta = O(\epsilon)$. The idea is to prove that, with overwhelming probability, there exists no solution to (76) such that $\epsilon = o(1)$. This then readily implies the claim of Theorem 2. To do so, we state and prove a number of preliminary results.

**Lemma C.1.** *We jointly have*

$$\|V\|_{\mathrm{op}} = O(\sqrt{p/d}), \qquad \|V\|_F^2 = O(p), \qquad \sum_{k=1}^p \|v_k\|_2^3 = O(p), \tag{85}$$

*with probability at least $1 - 2\exp\left(-c\log^2 d\right)$ over $V$. Furthermore, we have that*

$$\sup_{x \in \sqrt{d}\mathbb{S}^{d-1}} \|\varphi(x)\|_2 = \Theta(\sqrt{p}), \qquad \sup_{x \in \sqrt{d}\mathbb{S}^{d-1}} \|\tilde{\varphi}(x)\|_2 = \Theta(\sqrt{p}). \tag{86}$$

*with probability at least $1 - 2\exp\left(-c\log^2 d\right)$ over $V$.*

*Proof.* The first equation holds with probability at least $1 - 2\exp(-c_1 p)$ due to Theorem 4.4.5 in (Vershynin, 2018). The second equation is a direct consequence of Theorem 3.1.1 in (Vershynin, 2018). For the third equation, due to Theorem 3.1.1 in (Vershynin, 2018), we have that

$$\left\|\|v_k\|_2 - 1\right\|_{\psi_2} = O\left(\frac{1}{\sqrt{d}}\right), \tag{87}$$

which implies

$$\left\|\|v_k\|_2\right\|_{\psi_2} = O(1). \tag{88}$$

Then, we have that the Orlicz norm $\left\|\|v_k\|_2^3\right\|_{\psi_{2/3}} = O(1)$, which also implies $\mathbb{E}\left[\|v_k\|_2^3\right] = O(1)$. Then, due to Lemma B.6 in (Bombari et al., 2023), we have that

$$\sum_{k=1}^p \|v_k\|_2^3 = O(p), \tag{89}$$

with probability at least $1 - 2\exp\left(-c_1 \log^2 d\right)$, where we also used Assumption 3. The last statement statement can be obtained via the same argument in (31). $\square$

**Lemma C.2.** *Let $\rho_1, \rho_2, \rho_3 \in \mathbb{R}$ be sub-Gaussian random variables, not necessarily independent. Consider the random variable*

$$Z = \min(M, |\rho_1|)|\rho_2 \rho_3|, \tag{90}$$

*where $M = \omega(1)$. Then, $Z - \mathbb{E}[Z]$ is sub-exponential with parameters $(\nu, \alpha)$ (see Definition 2.7 in (Wainwright, 2019)) such that*

$$\nu = O(1), \qquad \alpha = O(M). \tag{91}$$

*Proof.* Since the $\rho$-s are sub-Gaussian, we have that both the absolute value of the mean and the second moment of $Z$ (and therefore its variance) are upper bounded by positive absolute constants. Denote with $C_1$ a constant order upper bound on the absolute value of $\mathbb{E}[Z]$. Furthermore, without loss of generality, we will consider the sub-Gaussian norms of $\rho_1, \rho_2, \rho_3$ to be equal to $1$.

Consider a real number $\lambda$ such that $4M|\lambda| \leq 1$. Defining $\bar{Z} = Z - \mathbb{E}[Z]$, and noting that $e^z \leq 1 + z + z^2 e^{|z|}/2$ for every $z \in \mathbb{R}$ (due to the mean-value theorem), we have that

$$
\begin{aligned}
\mathbb{E}\left[e^{\lambda \bar{Z}}\right] &\leq \mathbb{E}\left[1 + \lambda \bar{Z} + \frac{\lambda^2}{2}\bar{Z}^2 e^{|\lambda||\bar{Z}|}\right] \\
&\leq 1 + \mathbb{E}\left[\frac{\lambda^2}{2}\bar{Z}^2 e^{|\lambda||\bar{Z}|}\right] \\
&\leq 1 + \frac{\lambda^2}{2}\mathbb{E}\left[\bar{Z}^2 e^{\frac{M|\rho_2\rho_3|+C_1}{4M}}\right] \\
&\leq 1 + \frac{\lambda^2}{2}\mathbb{E}\left[(|\rho_1\rho_2\rho_3| + C_1)^2 e^{\frac{|\rho_2\rho_3|+1}{4}}\right] \\
&\leq 1 + \frac{\lambda^2}{2}\mathbb{E}\left[(|\rho_1\rho_2\rho_3| + C_1)^4\right]^{1/2} \mathbb{E}\left[e^{\frac{|\rho_2\rho_3|+1}{2}}\right]^{1/2} \\
&\leq 1 + C_2 \lambda^2 \\
&\leq e^{C_2 \lambda^2}.
\end{aligned}
\tag{92}
$$

Here, in third line we used $Z \leq M|\rho_2\rho_3|$; in the fourth line we used that $M \geq C_1$ and $Z \leq |\rho_1\rho_2\rho_3|$; in the sixth line we upper bounded the expectations via an absolute constant, as $|\rho_2\rho_3|$ is sub-exponential with norm $1$. Thus, we can set $\alpha = 4M$, and for all $|\lambda| \leq 1/\alpha$, we have

$$
\mathbb{E}\left[e^{\lambda \bar{Z}}\right] \leq e^{C_2 \lambda^2},
\tag{93}
$$

which gives the desired result according to Definition 2.7 in (Wainwright, 2019). $\square$

**Lemma C.3.** *We have that*

$$
\begin{aligned}
&\left|(\epsilon_2 - \epsilon_1)^\top V^\top \left((\varphi(x_1 + \epsilon_2) - \varphi(x_1 + \epsilon_1)) - \phi'(V(x_1 + \epsilon_1)) \circ (V(\epsilon_2 - \epsilon_1))\right)\right| \\
&\quad = O\left(p\delta^3 + d\log^2 d\delta^2\right),
\end{aligned}
\tag{94}
$$

*with probability at least $1 - 2\exp\left(-c_6 \log^2 d\right)$ over $V$.*

*Proof.* Since $\phi$ is differentiable due to Assumption 2, by the mean-value theorem, we have that there exists $\zeta \in \mathbb{R}^p$ such that

$$
\varphi(x_1 + \epsilon_2) - \varphi(x_1 + \epsilon_1) = \phi'(V(x_1 + \epsilon_1) + \zeta) \circ (V(\epsilon_2 - \epsilon_1)),
\tag{95}
$$

where each entry of $\zeta$ is such that $\zeta_k \in [0, [V(\epsilon_2 - \epsilon_1)]_k]$ (or $\zeta_k \in [[V(\epsilon_2 - \epsilon_1)]_k, 0]$, if $[V(\epsilon_2 - \epsilon_1)]_k$ is negative). Then, we have that the thesis becomes

$$
\begin{aligned}
&\left|(\epsilon_2 - \epsilon_1)^\top V^\top \left((\phi'(V(x_1 + \epsilon_1) + \zeta) - \phi'(V(x_1 + \epsilon_1))) \circ (V(\epsilon_2 - \epsilon_1))\right)\right| \\
&\quad = O\left(p\delta^3 + d\log^2 d\delta^2\right).
\end{aligned}
\tag{96}
$$

First, let $\xi \in \mathbb{R}^d$ be defined as

$$
\xi = \frac{\epsilon_2 - \epsilon_1}{\delta},
\tag{97}
$$

*i.e.* the vector lying on the sphere $\sqrt{d}\,\mathbb{S}^{d-1}$ with the same direction as $\epsilon_2 - \epsilon_1$. Then, dividing both sides of (96) by $\delta^3$ and expanding the sum, the desired result can be reformulated as

$$
\sum_{k=1}^{p} (v_k^\top \xi) \frac{\phi'(v_k^\top(x_1 + \epsilon_1) + \zeta_k) - \phi'(v_k^\top(x_1 + \epsilon_1))}{\delta} (v_k^\top \xi) = O\left(p + \frac{d\log^2 d}{\delta}\right).
\tag{98}
$$

Due to Assumption 2, we have both $\phi'(z) \leq L$ and $|\phi'(z_1) - \phi'(z_2)| \leq L'|z_1 - z_2|$. Thus, the previous equation is implied by

$$\sum_{k=1}^{p} \min\left(\frac{2L}{\delta}, L'\left|v_k^\top \xi\right|\right)\left(v_k^\top \xi\right)^2 = O\left(p + \frac{d\log^2 d}{\delta}\right), \tag{99}$$

as we have $|\zeta_k| \leq \delta|v_k^\top \xi|$ from its definition in (95).

Let us now consider an $\epsilon\sqrt{d}$-net of $\sqrt{d}\,\mathbb{S}^{d-1}$, namely $\{x_m^\epsilon\}_{m=1}^M$, such that for any $x \in \sqrt{d}\,\mathbb{S}^{d-1}$ there exists $m \in [M]$ such that $\|x - x_m^\epsilon\|_2 \leq \epsilon\sqrt{d}$. Due to Corollary 4.2.13 in (Vershynin, 2018), for $\epsilon < 1$ we have that the net can be chosen such that $M \leq (3/\epsilon)^d$. Setting $\epsilon = d^{-3/2}$, we have that there exists $m^* \in [M]$ such that $\|\xi - x_{m^*}^\epsilon\|_2 \leq 1/d$, and $M \leq e^{c_1 d\log d}$, where $c_1$ is an absolute positive constant. Consider a generic element $x_m^\epsilon$ and define

$$T^{(m)} = \sum_{k=1}^{p} \min\left(\frac{2L}{\delta}, L'\left|v_k^\top x_m^\epsilon\right|\right)\left(v_k^\top x_m^\epsilon\right)^2. \tag{100}$$

Each term $T_k^{(m)}$ in the sum above is such that its expectation is

$$\mathbb{E}_{v_k}\left[T_k^{(m)}\right] = \mathbb{E}_{v_k}\left[\min\left(\frac{2L}{\delta}, L'\left|v_k^\top x_m^\epsilon\right|\right)\left(v_k^\top x_m^\epsilon\right)^2\right] \leq L'\mathbb{E}_{v_k}\left[\left|v_k^\top x_m^\epsilon\right|^3\right] = O(1). \tag{101}$$

Furthermore, $T_k^{(m)}$ is sub-exponential (in the probability space of $v_k$) with parameters $(\nu, \alpha)$ (see Definition 2.7 in (Wainwright, 2019)) such that

$$\nu = O(1), \qquad \alpha = O\left(\delta^{-1}\right), \tag{102}$$

due to Lemma C.2. Thus, Equation (2.18) in (Wainwright, 2019) guarantees that

$$\mathbb{P}_V\left(\left|\sum_{k=1}^{p}\left(T_k^{(m)} - \mathbb{E}_{v_k}\left[T_k^{(m)}\right]\right)\right| \geq p + \frac{d\log^2 d}{\delta}\right) \leq \exp\left(-c_2 \min\left(p, d\log^2 d\right)\right) \\ \leq \exp\left(-c_3 d\log^2 d\right), \tag{103}$$

where the last step used Assumption 3. Next, we perform a union bound on the elements of the net, obtaining that

$$\sup_{m \in [M]}\left|T^{(m)}\right| = O\left(p + \frac{d\log^2 d}{\delta}\right), \tag{104}$$

with probability at least $1 - \exp\left(-c_3 d\log^2 d + c_1 d\log d\right) \geq 1 - 2\exp\left(-c_4 d\log^2 d\right)$. Then, with this same probability, due to (101), we also have that

$$\sum_{k=1}^{p} \min\left(\frac{2L}{\delta}, L'\left|v_k^\top x_{m^*}^\epsilon\right|\right)\left(v_k^\top x_{m^*}^\epsilon\right)^2 = O\left(p + \frac{d\log^2 d}{\delta}\right). \tag{105}$$

Since the $\min$ function is 1 Lipschitz in any of its arguments, for every $k \in [p]$, we have

$$\min\left(\frac{2L}{\delta}, L'\left|v_k^\top x_{m^*}^\epsilon\right|\right) - \min\left(\frac{2L}{\delta}, L'\left|v_k^\top \xi\right|\right) \leq L'\|v_k\|_2\|\xi - x_{m^*}^\epsilon\|_2 = O\left(\frac{\|v_k\|_2}{d}\right). \tag{106}$$

Thus,

$$\sum_{k=1}^{p}\left|\min\left(\frac{2L}{\delta}, L'\left|v_k^\top x_{m^*}^\epsilon\right|\right) - \min\left(\frac{2L}{\delta}, L'\left|v_k^\top \xi\right|\right)\right|\left(v_k^\top x_{m^*}^\epsilon\right)^2 \\ \leq \sum_{k=1}^{p}\frac{C_2}{d}\|v_k\|_2^3\|x_{m^*}^\epsilon\|_2^2 = O(p), \tag{107}$$

where the last step used the first statement of Lemma C.1, and holds with probability at least $1 - 2\exp\left(-c_5\log^2 d\right)$. Furthermore, for every $k \in [p]$, we have

$$\left|\left(v_k^\top x_{m^*}^\epsilon\right)^2 - \left(v_k^\top \xi\right)^2\right| = \left|v_k^\top\left(x_{m^*}^\epsilon - \xi\right)\right|\left|v_k^\top\left(x_{m^*}^\epsilon + \xi\right)\right| \\ \leq 2\sqrt{d}\|v_k\|_2^2\|x_{m^*}^\epsilon - \xi\|_2 = O\left(\frac{\|v_k\|_2^2}{\sqrt{d}}\right), \tag{108}$$

which yields

$$\sum_{k=1}^{p} \min\left(\frac{2L}{\delta}, L' \left|v_k^\top \xi\right|\right) \left|\left(v_k^\top x_{m^*}^\epsilon\right)^2 - \left(v_k^\top \xi\right)^2\right| \leq \sum_{k=1}^{p} \frac{C_3}{\sqrt{d}} \|v_k\|_2^3 \|\xi\|_2 = O(p), \qquad (109)$$

again due to Lemma C.1. Finally, applying the triangle inequality to (105), (107), and (109), gives

$$\sum_{k=1}^{p} \min\left(\frac{2L}{\delta}, L' \left|v_k^\top \xi\right|\right) \left(v_k^\top \xi\right)^2 = O\left(p + \frac{d \log^2 d}{\delta}\right), \qquad (110)$$

with probability at least $1 - 2\exp\left(-c_6 \log^2 d\right)$, which concludes the proof. $\qquad \square$

**Lemma C.4.** *Suppose $\epsilon = o(1)$. Then, we have that*

$$\left|(\epsilon_2 - \epsilon_1)^\top V^\top \left(\phi'(V(x_1 + \epsilon_1)) \circ (V(\epsilon_2 - \epsilon_1))\right) - \frac{\|\epsilon_2 - \epsilon_1\|_2^2}{d} p\mu_1\right| = o(p\delta^2), \qquad (111)$$

*with probability at least $1 - 2\exp\left(-c \log^2 d\right)$ over $V$.*

*Proof.* First, let $\xi \in \mathbb{R}^d$ be defined as in (97). Then, the desired result can be reformulated as

$$\left|\sum_{k=1}^{p} \phi'\left(v_k^\top (x_1 + \epsilon_1)\right) \left(v_k^\top \xi\right)^2 - p\mu_1\right| = o(p). \qquad (112)$$

Let us now consider an $\epsilon \sqrt{d}$-net of $\sqrt{d}\, \mathbb{S}^{d-1}$ (as we did after (99)). Setting $\epsilon = d^{-2}$, we have that there exist $m^{(\xi)}$ and $m^{(x)}$ in $[M]$ such that $\left\|\xi - x_{m^{(\xi)}}^\epsilon\right\|_2 \leq 1/d^{3/2}$, $\left\|(x_1 + \epsilon_1) - x_{m^{(x)}}^\epsilon\right\|_2 \leq 1/d^{3/2}$, and the size of the net is $M \leq e^{c_1 d \log d}$, where $c_1$ is an absolute positive constant. Consider two generic elements of this net: $x_{m^{(1)}}^\epsilon$ and $x_{m^{(2)}}^\epsilon$, and define

$$T^{(m^{(1)}, m^{(2)})} = \left|\sum_{k=1}^{p} \left(\phi'\left(v_k^\top x_{m^{(1)}}^\epsilon\right) \left(v_k^\top x_{m^{(2)}}^\epsilon\right)^2 - \mathbb{E}_{v_k}\left[\phi'\left(v_k^\top x_{m^{(1)}}^\epsilon\right) \left(v_k^\top x_{m^{(2)}}^\epsilon\right)^2\right]\right)\right|. \qquad (113)$$

Since $\left|\phi'\left(v_k^\top x_{m^{(1)}}^\epsilon\right)\right| \leq L$, each element of the sum is sub-exponential (in the probability space of $v_k$). Then, due to Bernstein inequality (see Theorem 2.8.1 in (Vershynin, 2018)), we have that

$$\mathbb{P}_V\left(T^{(m^{(1)}, m^{(2)})} > C_1 \sqrt{dp \log d}\right) \leq 2\exp\left(-c_2 \min\left(C_1^2 d \log d, C_1 \sqrt{dp \log d}\right)\right)$$
$$\leq 2\exp\left(-c_3 C_1 d \log d\right), \qquad (114)$$

where the second step is a consequence of Assumption 3. Performing a union bound, we have that, for every pair of points on the net (there are less than $e^{2c_1 d \log d}$ such pairs), including $x_{m^{(\xi)}}^\epsilon$ and $x_{m^{(x)}}^\epsilon$, the equation above holds with probability at least $1 - 2\exp\left(-c_3 C_1 d \log d + 2c_1 d \log d\right) \geq 1 - 2\exp\left(-c_4 d \log d\right)$ if $C_1$ is chosen sufficiently large. Thus, with this probability, we have

$$T^{(m^{(x)}, m^{(\xi)})} = \left|\sum_{k=1}^{p} \left(\phi'\left(v_k^\top x_{m^{(x)}}^\epsilon\right) \left(v_k^\top x_{m^{(\xi)}}^\epsilon\right)^2 - \mathbb{E}_{v_k}\left[\phi'\left(v_k^\top x_{m^{(x)}}^\epsilon\right) \left(v_k^\top x_{m^{(\xi)}}^\epsilon\right)^2\right]\right)\right|$$
$$= \left|\sum_{k=1}^{p} \phi'\left(v_k^\top x_{m^{(x)}}^\epsilon\right) \left(v_k^\top x_{m^{(\xi)}}^\epsilon\right)^2 - p\,\mathbb{E}\left[\phi'\left(\rho_x^\epsilon\right) \left(\rho_\xi^\epsilon\right)^2\right]\right| \qquad (115)$$
$$= O\left(\sqrt{dp \log d}\right) = o(p),$$

where we used Assumption 3 and introduced $\rho_x^\epsilon$ and $\rho_\xi^\epsilon$ as two standard Gaussian variables with correlation $\left(x_{m^{(x)}}^\epsilon\right)^\top x_{m^{(\xi)}}^\epsilon / d$. Note that

$$\left(\rho_\xi^\epsilon\right)^2 = 1 + \sqrt{2}\frac{\left(\rho_\xi^\epsilon\right)^2 - 1}{\sqrt{2}} = h_0\left(\rho_\xi^\epsilon\right) + \sqrt{2}h_2\left(\rho_\xi^\epsilon\right), \qquad (116)$$

where $h_0$ and $h_2$ denote the 0th and 2nd Hermite polynomials. Thus, we have that

$$\mathbb{E}\left[\phi'\left(\rho_x^\epsilon\right)\left(\rho_\xi^\epsilon\right)^2\right] = \mu_0^{\phi'} + \sqrt{2}\mu_2^{\phi'}\left(\frac{\left(x_{m^{(x)}}^\epsilon\right)^\top x_{m^{(\xi)}}^\epsilon}{d}\right)^2, \tag{117}$$

where $\mu_0^{\phi'}$ is the 0th Hermite coefficient of $\phi'$. Note that $\mu_0^{\phi'}$ corresponds to the 1st Hermite coefficient of $\phi$, which is $\mu_1 \neq 0$ due to Assumption 2. The second term on the RHS of (117) can be bounded via

$$\begin{aligned}
\left|\left(x_{m^{(x)}}^\epsilon\right)^\top x_{m^{(\xi)}}^\epsilon\right| &\leq \left\|x_{m^{(x)}}^\epsilon\right\|_2\left\|x_{m^{(\xi)}}^\epsilon - \xi\right\|_2 + \left\|x_{m^{(x)}}^\epsilon - x_1 - \epsilon_1\right\|_2\|\xi\|_2 + \left|(x_1+\epsilon_1)^\top\xi\right| \\
&\leq \frac{2}{d} + \left|\frac{x_1^\top(\epsilon_1-\epsilon_2)}{\delta}\right| + \left|\frac{\epsilon_1^\top(\epsilon_1-\epsilon_2)}{\delta}\right| = O\left(\frac{1}{d} + d\epsilon\right),
\end{aligned} \tag{118}$$

where we used (82) in the last step. Then, plugging in (117), we obtain

$$\left|\mathbb{E}\left[\phi'\left(\rho_x^\epsilon\right)\left(\rho_\xi^\epsilon\right)^2\right] - \mu_1\right| = O\left(\frac{1}{d^4} + \epsilon^2\right) = o(1). \tag{119}$$

We also have

$$\begin{aligned}
&\left|\phi'\left(v_k^\top x_{m^{(x)}}^\epsilon\right)\left(v_k^\top x_{m^{(\xi)}}^\epsilon\right)^2 - \phi'\left(v_k^\top(x_1+\epsilon_1)\right)\left(v_k^\top\xi\right)^2\right| \\
&\leq \left|\left(\phi'\left(v_k^\top x_{m^{(x)}}^\epsilon\right) - \phi'\left(v_k^\top(x_1+\epsilon_1)\right)\right)\left(v_k^\top x_{m^{(\xi)}}^\epsilon\right)^2\right| \\
&\quad + \left|\phi'\left(v_k^\top(x_1+\epsilon_1)\right)\left(\left(v_k^\top x_{m^{(\xi)}}^\epsilon\right)^2 - \left(v_k^\top\xi\right)^2\right)\right| \\
&\leq L'\|v_k\|_2\left\|x_{m^{(x)}}^\epsilon - x_1 - \epsilon_1\right\|_2\|v_k\|_2^2 d + 2L\sqrt{d}\|v_k\|_2^2\left\|x_{m^{(\xi)}}^\epsilon - \xi\right\|_2 \\
&\leq C_2\frac{\|v_k\|_2^2 + \|v_k\|_2^3}{\sqrt{d}},
\end{aligned} \tag{120}$$

where $C_2$ is some positive constant. This yields

$$\sum_{k=1}^p\left|\phi'\left(v_k^\top x_{m^{(x)}}^\epsilon\right)\left(v_k^\top x_{m^{(\xi)}}^\epsilon\right)^2 - \phi'\left(v_k^\top(x_1+\epsilon_1)\right)\left(v_k^\top\xi\right)^2\right| = O\left(\frac{p}{\sqrt{d}}\right) = o(p), \tag{121}$$

with probability at least $1 - 2\exp\left(-c_5\log^2 d\right)$ due to Lemma C.1. Then, applying the triangle inequality to (115), (119), and (121), the proof is complete. $\qquad\square$

**Lemma C.5.** *Suppose $\epsilon = o(1)$. Then, we have that*

$$|a_1 + a_2| = O\left(|a_2|\delta + \frac{\log d}{\sqrt{d}}\right), \tag{122}$$

*with probability at least $1 - 2\exp\left(-c\log^2 d\right)$ over $V$ and $X$.*

*Proof.* Consider (78), written as

$$\varphi(x_2) = (a_1+a_2)\varphi(x_1+\epsilon_1) + a_2\left(\varphi(x_1+\epsilon_2) - \varphi(x_1+\epsilon_1)\right). \tag{123}$$

Let us now take an inner product of both sides of (123) with $V x_1$. Due to Lemma C.1 and since $x_1$ is a sub-Gaussian vector independent of $V$ and $x_2$ due to Assumption 1, we have that

$$\left|x_1^\top V^\top\varphi(x_2)\right| = O\left(\frac{p}{\sqrt{d}}\log d\right), \tag{124}$$

with probability at least $1 - 2\exp\left(-c_1\log^2 d\right)$ over $x_1$ and $V$. Then, consider

$$\begin{aligned}
\left|x_1^\top V^\top\varphi(x_1+\epsilon_1) - \mu_1 p\right| &\leq \left|x_1^\top V^\top\phi\left(V(x_1+\epsilon_1)\right) - x_1^\top V^\top\phi(Vx_1)\right| \\
&\quad + \left|x_1^\top V^\top\phi(Vx_1) - \mu_1 p\right|,
\end{aligned} \tag{125}$$

and let us bound the two terms on the RHS separately. For the first one, since $\phi$ is $L$-Lipschitz by Assumption 2, we have that

$$\left|x_1^\top V^\top \phi\left(V(x_1 + \epsilon_1)\right) - x_1^\top V^\top \phi(Vx_1)\right| \le L \left\|Vx_1\right\|_2 \left\|V\epsilon_1\right\|_2 = O\left(p\epsilon\right), \tag{126}$$

with probability at least $1 - 2\exp\left(-c_1 \log^2 d\right)$ due to Lemma C.1 and (83). For the second term in the RHS of (125), using the Hermite decomposition of $\phi$, we have that

$$\left|x_1^\top V^\top \phi(Vx_1) - \mu_1 p\right| = \left|\sum_{k=1}^p v_k^\top x_1 \phi(v_k^\top x_1) - \mathbb{E}_{v_k}\left[v_k^\top x_1 \phi(v_k^\top x_1)\right]\right|. \tag{127}$$

Since $\phi$ is Lipschitz, we have that $\phi(v_k^\top x_1)$ is a sub-Gaussian random variable (with respect to $v_k$), and thus $v_k^\top x_1 \phi(v_k^\top x_1)$ are $p$ i.i.d. sub-exponential random variables. Thus, Bernstein inequality (see Theorem 2.8.1. in (Vershynin, 2018)) yields

$$\left|x_1^\top V^\top \phi(Vx_1) - \mu_1 p\right| = O\left(\sqrt{p} \log d\right) \tag{128}$$

with probability at least $1 - 2\exp\left(-c_3 \log^2 d\right)$ over $V$. Then, plugging this and (126) in (125), together with the fact that $\mu_1 \ne 0$ by Assumption 2, we have

$$\left|x_1^\top V^\top \varphi(x_1 + \epsilon_1) - \mu_1 p\right| = O\left(p\epsilon + \sqrt{p} \log d\right) = o(p), \tag{129}$$

with probability at least $1 - 2\exp\left(-c_4 \log^2 d\right)$ over $V$. Considering now the last term in (123), we have

$$\left\|\varphi(x_1 + \epsilon_2) - \varphi(x_1 + \epsilon_1)\right\|_2 = O\left(\sqrt{p}\delta\right), \tag{130}$$

due to the Lipschitzness of $\phi$ and due to Lemma C.1. Thus, with probability $1 - 2\exp\left(-c_5 \log^2 d\right)$, we have

$$\left|x_1^\top V^\top \left(\varphi(x_1 + \epsilon_2) - \varphi(x_1 + \epsilon_1)\right)\right| = O\left(p\delta\right), \tag{131}$$

where we used again Lemma C.1. Then, plugging (124), (129) and (131) in (123), an application of the triangle inequality yields

$$|a_1 + a_2| p = O\left(|a_2| p\delta + \frac{p \log d}{\sqrt{d}}\right), \tag{132}$$

which gives the desired result. □

**Lemma C.6.** *Suppose $\epsilon = o(1)$. Then, we have that*

$$|a_2| = O(\delta^{-1}), \qquad |a_1 + a_2| = O(1), \tag{133}$$

*with probability at least $1 - 2\exp\left(-c \log^2 d\right)$ over $V$ and $X$.*

*Proof.* The proof consists in considering the inner product of both sides of (78), namely

$$\begin{aligned}
\varphi(x_2) = {}& (a_1 + a_2)\varphi(x_1 + \epsilon_1) \\
& + a_2\left(\left(\varphi(x_1 + \epsilon_2) - \varphi(x_1 + \epsilon_1)\right) - \phi'(V(x_1 + \epsilon_1)) \circ (V(\epsilon_2 - \epsilon_1))\right) \\
& + a_2\, \phi'(V(x_1 + \epsilon_1)) \circ (V(\epsilon_2 - \epsilon_1)),
\end{aligned} \tag{134}$$

with $V(\epsilon_2 - \epsilon_1)$.

First, we have that the LHS reads

$$\left|(\epsilon_2 - \epsilon_1)^\top V^\top \varphi(x_2)\right| = O\left(\delta p\right), \tag{135}$$

due to Lemma C.1 with probability at least $1 - 2\exp\left(-c_1 \log^2 d\right)$ over $V$. Consider now $(\epsilon_2 - \epsilon_1)^\top V^\top \varphi(x_1 + \epsilon_1)$. A net argument similar to the one used to obtain the third statement in Lemma B.1 yields

$$\left|(\epsilon_2 - \epsilon_1)^\top V^\top \varphi(x_1 + \epsilon_1) - \mu_1 p \frac{(\epsilon_2 - \epsilon_1)^\top (x_1 + \epsilon_1)}{d}\right| = O\left(\delta\sqrt{pd} \log d + \frac{\delta p}{d^2}\right), \tag{136}$$

with probability at least $1 - 2\exp\left(-c_2 \log^2 d\right)$ over $V$. This, together with (82) and Lemma C.5, gives

$$\left|(a_1 + a_2)(\epsilon_2 - \epsilon_1)^\top V^\top \varphi(x_1 + \epsilon_1)\right| = O\left(\left(|a_2|\delta + \frac{\log d}{\sqrt{d}}\right)\left(\delta\sqrt{pd}\log d + \frac{\delta p}{d^2} + \delta\epsilon p\right)\right), \tag{137}$$

with probability at least $1 - 2\exp\left(-c_3 \log^2 d\right)$ over $V$ and $X$. Due to Lemma C.3, we have

$$\begin{aligned}
&\left|(\epsilon_2 - \epsilon_1)^\top V^\top \left((\varphi(x_1 + \epsilon_2) - \varphi(x_1 + \epsilon_1)) - \phi'(V(x_1 + \epsilon_1)) \circ (V(\epsilon_2 - \epsilon_1))\right)\right| \\
&\qquad = O\left(p\delta^3 + d\log^2 d\delta^2\right) = o(p\delta^2),
\end{aligned} \tag{138}$$

and due to Lemma C.4, we have

$$\left|(\epsilon_2 - \epsilon_1)^\top V^\top \left(\phi'(V(x_1 + \epsilon_1)) \circ (V(\epsilon_2 - \epsilon_1))\right) - \frac{\|\epsilon_2 - \epsilon_1\|_2^2}{d} p\mu_1\right| = o(p\delta^2), \tag{139}$$

jointly with probability $1 - 2\exp\left(-c_4 \log^2 d\right)$ over $V$. Then, taking (135), (137), (138), and (139) together, we have that

$$|a_2|\frac{\|\epsilon_2 - \epsilon_1\|_2^2}{d} p\mu_1 = A_1 + A_2 + A_3 + A_4, \tag{140}$$

where

$$\begin{aligned}
|A_1| &= O(\delta p), \\
|A_2| &= O\left(\left(|a_2|\delta + \frac{\log d}{\sqrt{d}}\right)\left(\delta\sqrt{pd}\log d + \frac{\delta p}{d^2} + \delta\epsilon p\right)\right) = o(\delta p) + o\left(|a_2|p\delta^2\right), \\
|A_3| &= o\left(|a_2|p\delta^2\right), \\
|A_4| &= o\left(|a_2|p\delta^2\right).
\end{aligned} \tag{141}$$

As $\mu_1 \neq 0$, this readily implies that

$$|a_2| = O\left(\delta^{-1}\right), \tag{142}$$

which, together with Lemma C.5 gives the desired result. $\qquad\square$

**Lemma C.7.** *Suppose $\epsilon = o(1)$. Then, we have that*

$$\left|a_2\tilde{\varphi}(x_2)^\top \left(\varphi(x_1 + \epsilon_2) - \varphi(x_1 + \epsilon_1)\right)\right| = o(p). \tag{143}$$

*with probability at least $1 - 2\exp\left(-c\log^2 d\right)$ over $V$ and $X$.*

*Proof.* First, following the same decomposition considered in (95), we have

$$\begin{aligned}
&\left|a_2\tilde{\varphi}(x_2)^\top \left(\varphi(x_1 + \epsilon_2) - \varphi(x_1 + \epsilon_1)\right)\right| \\
&\leq \left|a_2\tilde{\varphi}(x_2)^\top \left((\phi'(V(x_1 + \epsilon_1) + \zeta) - \phi'(V(x_1 + \epsilon_1))) \circ (V(\epsilon_2 - \epsilon_1))\right)\right| \\
&\quad + \left|a_2\tilde{\varphi}(x_2)^\top \phi'(V(x_1 + \epsilon_1)) \circ (V(\epsilon_2 - \epsilon_1))\right|,
\end{aligned} \tag{144}$$

where each entry of $\zeta$ is such that $\zeta_k \in [0, [V(\epsilon_2 - \epsilon_1)]_k]$ (or $\zeta_k \in [[V(\epsilon_2 - \epsilon_1)]_k, 0]$, if $[V(\epsilon_2 - \epsilon_1)]_k$ is negative). The first term on the RHS can be bounded as

$$\begin{aligned}
&\left|a_2\tilde{\varphi}(x_2)^\top \left((\phi'(V(x_1 + \epsilon_1) + \zeta) - \phi'(V(x_1 + \epsilon_1))) \circ (V(\epsilon_2 - \epsilon_1))\right)\right| \\
&\qquad \leq |a_2|\sum_{k=1}^p \left|\tilde{\phi}(v_k^\top x_2)\right|\min\left(2L, L'\left|v_k^\top(\epsilon_2 - \epsilon_1)\right|\right)\left|v_k^\top(\epsilon_2 - \epsilon_1)\right| \\
&\qquad = |a_2|O\left(p\delta^2 + d\log^2 d\delta\right) = O\left(p\delta + d\log^2 d\right),
\end{aligned} \tag{145}$$

with probability at least $1 - 2\exp\left(-c_1 \log^2 d\right)$ over $V$ and $X$, due to the same argument used in Lemma C.3 and where we used also Lemma C.6.

Next, let us look at the second term on the RHS of (144), and consider an $\epsilon\sqrt{d}$-net of $\sqrt{d}\,\mathbb{S}^{d-1}$ (as we did after (99)). Setting $\epsilon = d^{-2}$, we have that there exist $m^{(\xi)}$, $m^{(x_1)}$ and $m^{(x_2)}$ in $[M]$ such that $\left\|\xi - x_{m^{(\xi)}}^\epsilon\right\|_2 \leq 1/d^{3/2}$ (where $\xi$ is defined in (97)), $\left\|(x_1 + \epsilon_1) - x_{m^{(x_1)}}^\epsilon\right\|_2 \leq 1/d^{3/2}$,

$\left\|(x_2) - x^\epsilon_{m^{(x_2)}}\right\|_2 \leq 1/d^{3/2}$, and the size of the net is $M \leq e^{c_2 d \log d}$, where $c_2$ is an absolute positive constant. Consider three generic elements of this net: $x^\epsilon_{m^{(1)}}$, $x^\epsilon_{m^{(2)}}$, and $x^\epsilon_{m^{(3)}}$, and define

$$
T^{(m^{(1)}, m^{(2)}, m^{(3)})} = \left| \sum_{k=1}^{p} \left( \tilde{\phi}(v_k^\top x^\epsilon_{m^{(1)}}) \phi'\left(v_k^\top x^\epsilon_{m^{(2)}}\right) \left(v_k^\top x^\epsilon_{m^{(3)}}\right) \right. \right.
$$
$$
\left. \left. - \mathbb{E}_{v_k} \left[ \tilde{\phi}(v_k^\top x^\epsilon_{m^{(1)}}) \phi'\left(v_k^\top x^\epsilon_{m^{(2)}}\right) \left(v_k^\top x^\epsilon_{m^{(3)}}\right) \right] \right) \right|. \tag{146}
$$

Since $\left| \phi'\left(v_k^\top x^\epsilon_{m^{(1)}}\right) \right| \leq L$, each element of the sum is sub-exponential (in the probability space of $v_k$). Then, the argument based on Bernstein inequality used in Lemma C.4, after performing a union bound on the net, yields

$$
T^{(m^{(x_2)}, m^{(x_1)}, m^{(\xi)})} =
$$
$$
\left| \sum_{k=1}^{p} \left( \tilde{\phi}(v_k^\top x^\epsilon_{m^{(x_2)}}) \phi'\left(v_k^\top x^\epsilon_{m^{(x_1)}}\right) \left(v_k^\top x^\epsilon_{m^{(\xi)}}\right) \right) - p\, \mathbb{E}\left[ \tilde{\phi}\left(\rho^\epsilon_{x_2}\right) \phi'\left(\rho^\epsilon_{x_1}\right) \left(\rho^\epsilon_\xi\right) \right] \right| = o(p), \tag{147}
$$

with probability at least $1 - 2\exp\left(-c_3 d \log^2 d\right)$ (due to Assumption 3), where we introduced $\rho^\epsilon_{x_1}$, $\rho^\epsilon_{x_2}$ and $\rho^\epsilon_\xi$ as three standard Gaussian variables with correlations

$$
\rho_{12} := \text{corr}\left(\rho^\epsilon_{x_1}, \rho^\epsilon_{x_2}\right) = \frac{\left(x^\epsilon_{m^{(x_1)}}\right)^\top x^\epsilon_{m^{(x_2)}}}{d}, \qquad \rho_{1\xi} := \text{corr}\left(\rho^\epsilon_{x_1}, \rho^\epsilon_\xi\right) = \frac{\left(x^\epsilon_{m^{(x_1)}}\right)^\top x^\epsilon_{m^{(\xi)}}}{d},
$$
$$
\rho_{2\xi} := \text{corr}\left(\rho^\epsilon_{x_2}, \rho^\epsilon_\xi\right) = \frac{\left(x^\epsilon_{m^{(x_2)}}\right)^\top x^\epsilon_{m^{(\xi)}}}{d}. \tag{148}
$$

Note that

$$
|\rho_{12}| = o(1), \qquad |\rho_{1\xi}| = o(1), \tag{149}
$$

where the two bounds hold due to our net definition, due to $|x_1^\top x_2| = o(d)$ with probability at least $1 - 2\exp(-c_4 \log^2 d)$ (coming from Assumption 1), and due to (82). By Isserlis' theorem (or generalized Stein's lemma) we also have

$$
\mathbb{E}\left[ \tilde{\phi}(\rho^\epsilon_{x_2}) \phi'\left(\rho^\epsilon_{x_1}\right) \rho^\epsilon_\xi \right] = \rho_{2\xi} \mathbb{E}\left[ \tilde{\phi}'(\rho^\epsilon_{x_2}) \phi'\left(\rho^\epsilon_{x_1}\right) \right] + \rho_{1\xi} \mathbb{E}\left[ \tilde{\phi}(\rho^\epsilon_{x_2}) \phi''\left(\rho^\epsilon_{x_1}\right) \right]. \tag{150}
$$

We have that

$$
\left| \rho_{1\xi} \mathbb{E}\left[ \tilde{\phi}(\rho^\epsilon_{x_2}) \phi''\left(\rho^\epsilon_{x_1}\right) \right] \right| \leq |\rho_{1\xi}| \mathbb{E}\left[ \tilde{\phi}(\rho^\epsilon_{x_2})^2 \right]^{1/2} \mathbb{E}\left[ \phi''\left(\rho^\epsilon_{x_1}\right)^2 \right]^{1/2} = o(1), \tag{151}
$$

where we used (149), $|\phi''| \leq L'$ ($L'$ being the Lipschitz constant of $\phi'$) and that $\tilde{\phi}$ is Lipschitz due to Assumption 2. To study the first term on the LHS of (150), notice that the Hermite coefficient of order 0 of $\tilde{\phi}'$ is 0, since the Hermite coefficient of order 1 of $\tilde{\phi}$ is 0 by definition. Thus, denoting by $\mu'_r, \tilde{\mu}'_r$ the $r$-th Hermite coefficient respectively of $\phi', \tilde{\phi}'$, we have

$$
\left| \rho_{2\epsilon} \mathbb{E}\left[ \tilde{\phi}'(\rho^\epsilon_{x_2}) \phi'\left(\rho^\epsilon_{x_1}\right) \right] \right| \leq |\rho_{2\epsilon}| \left| \sum_{r=1}^{\infty} \tilde{\mu}'_r \mu'_r \rho_{12}^r \right| \leq |\rho_{12}| \sum_{r=1}^{\infty} |\tilde{\mu}'_r \mu'_r|
$$
$$
\leq |\rho_{12}| \sqrt{\sum_{r=1}^{\infty} (\tilde{\mu}'_r)^2} \sqrt{\sum_{r=1}^{\infty} (\mu'_r)^2} \tag{152}
$$
$$
\leq |\rho_{12}| \mathbb{E}\left[ \tilde{\phi}'(\rho^\epsilon_{x_2})^2 \right]^{1/2} \mathbb{E}\left[ \phi'\left(\rho^\epsilon_{x_1}\right)^2 \right]^{1/2} = o(1),
$$

where in the last step we used (149), $|\phi'| \leq L$ and that $\tilde{\phi}$ is Lipschitz due to Assumption 2. Putting (151) and (152) in (150), and plugging this in (147) yields

$$
\left| \sum_{k=1}^{p} \tilde{\phi}(v_k^\top x^\epsilon_{m^{(x_2)}}) \phi'\left(v_k^\top x^\epsilon_{m^{(x_1)}}\right) \left(v_k^\top x^\epsilon_{m^{(\xi)}}\right) \right| = o(p), \tag{153}
$$

with probability at least $1 - 2\exp(-c_5 \log^2 d)$ over $V$ and $X$.

Then, following a similar argument as the one that led to (121), we obtain

$$\left| \sum_{k=1}^{p} \tilde{\phi}(v_k^\top x_{m(x_2)}^\epsilon) \phi'\left(v_k^\top x_{m(x_1)}^\epsilon\right)\left(v_k^\top x_{m(\xi)}^\epsilon\right) - \tilde{\phi}(v_k^\top x_2)\phi'\left(v_k^\top(x_1+\epsilon_1)\right)\left(v_k^\top \xi\right)\right| = o(p), \quad (154)$$

with probability at least $1 - 2\exp\left(-c_6 \log^2 d\right)$ over $V$ due to Lemma C.1. Putting (153) and (154) together with the result of Lemma C.6 yields

$$\left| a_2\tilde{\varphi}(x_2)^\top \phi'(V(x_1+\epsilon_1)) \circ (V(\epsilon_2 - \epsilon_1))\right| = o(p), \quad (155)$$

with probability at least $1 - 2\exp\left(-c_7 \log^2 d\right)$ over $V$ and $X$. This last equation, when plugged in (144) together with (145), gives the desired result. $\qquad\square$

**Proof of Theorem 2.** Let us consider

$$\varphi(x_2) = a_1\varphi(x_1+\epsilon_1) + a_2\varphi(x_1+\epsilon_2), \quad (156)$$

where

$$\|x_1+\epsilon_1\|_2 = \|x_1+\epsilon_2\|_2 = \sqrt{d}. \quad (157)$$

Suppose by contradiction that there exists a solution to the equations above such that $\epsilon = o(1)$. Take the inner product of both sides of (156) with the vector $\tilde{\varphi}(x_2)$, namely

$$\tilde{\varphi}(x_2)^\top \varphi(x_2) = (a_1+a_2)\tilde{\varphi}(x_2)^\top \varphi(x_1+\epsilon_1) + a_2\tilde{\varphi}(x_2)^\top\left(\varphi(x_1+\epsilon_2) - \varphi(x_1+\epsilon_1)\right). \quad (158)$$

For the LHS, we have that

$$\tilde{\varphi}(x_2)^\top \varphi(x_2) = \Theta(p), \quad (159)$$

with probability at least $1 - 2\exp\left(-c_1 \log^2 d\right)$, due to the first statement of Lemma B.1. For the first term of the RHS, we have

$$\left|(a_1+a_2)\tilde{\varphi}(x_2)^\top \varphi(x_1+\epsilon_1) - (a_1+a_2)\tilde{\varphi}(x_2)^\top \varphi(x_1)\right|$$
$$\leq |a_1+a_2|\,\|\tilde{\varphi}(x_2)\|_2\, L\, \|V\|_{\mathrm{op}}\, \|\epsilon_1\|_2 = o(p), \quad (160)$$

with probability at least $1 - 2\exp\left(-c_2 \log^2 d\right)$, due to Lemmas C.1 and C.6. Then, using the Hermite expansion of $\tilde{\phi}$ and $\phi$ we have

$$\left| \tilde{\varphi}(x_2)^\top \varphi(x_1) - p\sum_{r=3}^{\infty}\left(\frac{x_2^\top x_1}{d}\right)^r\right| = o(p), \quad (161)$$

with probability at least $1 - 2\exp\left(-c_3 \log^2 d\right)$ due to Bernstein inequality. As we have $|x_2^\top x_1| = o(d)$ with this same probability due to Assumption 1, we readily obtain

$$\left|(a_1+a_2)\tilde{\varphi}(x_2)^\top \varphi(x_1)\right| = o(p), \quad (162)$$

where we also used the bound on $|a_1+a_2|$ from Lemma C.6.

Finally, due to Lemma C.7, we have

$$\left|a_2\tilde{\varphi}(x_2)^\top\left(\varphi(x_1+\epsilon_2) - \varphi(x_1+\epsilon_1)\right)\right| = o(p) \quad (163)$$

with probability at least $1 - 2\exp\left(-c_4 \log^2 d\right)$.

Plugging (159), (160), (162) and (163) in (158) generates a contradiction with high probability. This implies that, with probability at least $1 - 2\exp\left(-c_5 \log^2 d\right)$, we have $\epsilon = \Omega(1)$. Taking the intersection between this event and the one described by Theorem 1, we obtain the thesis. $\qquad\square$

# D   Implementation details

**Computational resources.** We executed all the experiments on a machine equipped with two GPUs (an NVIDIA GeForce RTX 3090, 24 GB VRAM and an NVIDIA RTX A5000, 24 GB VRAM), an Intel(R) Core(TM) i9-10920X CPU @ 3.50GHz and 128 GB of RAM.

**Training procedure of neural networks.** By default, we train both two-layer and deep residual networks with full-batch gradient descent with step size $10^{-5}$ on square loss for $10^{6}$ steps, unless stated otherwise. For the classification experiments on random features and two-layer neural networks, we set the step size to $10^{-3}$ and optimize logistic loss (in the case of binary classification) or cross-entropy loss (in the case of multi-class classification) with full-batch gradient descent for $10^{6}$ steps.

**Optimization details for the reconstruction algorithm.** We provide precise implementation details of the reconstruction algorithm. The inputs of our procedure are the model's optimal weights $\theta^*$ and the structure of $f_{\text{RF}}$, *i.e.*, $V$ and $\phi$. As already mentioned in Section 5.2, in the case of neural networks where initialization is not zero, we define $\theta^* \triangleq \theta_*^{(L)} - \theta_0^{(L)}$, where $\theta_0$ are the initial parameters and $\theta^*$ are the trained parameters after gradient descent converged on the training set. The superscript $^{(L)}$ indicates the weights of the last layer. For neural networks we also assume access to their trained weights of all layers $\{\theta^{*(1)}, \ldots, \theta^{*(L)}\}$, but we only require knowledge of the initialization of the last layer $\theta_0^{(L)}$ and their internal structure. The reconstruction problem is solved by minimizing the following objective via gradient descent with momentum:

$$\hat{X}^\star = \underset{\hat{X}:\|\hat{x}_i\|_2=\sqrt{d}}{\arg\min} \left\| P_{\hat{\Phi}}^{\perp}\theta^* \right\|_2^2. \tag{164}$$

We initialize the rows $\hat{x}_i$ as i.i.d. standard Gaussian vectors. To efficiently compute the objective in (164) at each gradient descent iterate, we leverage the fact that, by definition, $P_{\hat{\Phi}}^{\perp}$ is symmetric $((P_{\hat{\Phi}}^{\perp})^\top = P_{\hat{\Phi}}^{\perp})$ and idempotent $((P_{\hat{\Phi}}^{\perp})^2 = P_{\hat{\Phi}}^{\perp})$. Expanding according to the definitions and properties above,

$$\begin{aligned}
\mathcal{L}(\hat{X}) = \left\| P_{\hat{\Phi}}^{\perp}\theta^* \right\|_2^2 &= [P_{\hat{\Phi}}^{\perp}\theta^*]^\top P_{\hat{\Phi}}^{\perp}\theta^* \\
&= [\theta^*]^\top [P_{\hat{\Phi}}^{\perp}]^\top P_{\hat{\Phi}}^{\perp}\theta^* \\
&= [\theta^*]^\top P_{\hat{\Phi}}^{\perp}\theta^* \\
&= [\theta^*]^\top (I - \hat{\Phi}^+\hat{\Phi})\theta^* \\
&= [\theta^*]^\top \theta^* - [\theta^*]^\top \hat{\Phi}^\top (\hat{\Phi}\hat{\Phi}^\top)^{-1}\hat{\Phi}\theta^* \\
&= [\theta^*]^\top \theta^* - [\theta^*]^\top \hat{\Phi}^\top \alpha \,.
\end{aligned}$$

Here, $\alpha$ is the solution of the system of $n$ linear equations in $n$ unknowns $(\hat{\Phi}\hat{\Phi}^\top)\,\alpha = \hat{\Phi}\theta^*$ which can be numerically computed via the conjugate gradient method (Hestenes et al., 1952), skipping the need of inverting explicitly $\hat{\Phi}\hat{\Phi}^\top \in \mathbb{R}^{n\times n}$ at each iteration. After the gradient descent update on $\hat{X}$, we then normalize its rows $\hat{x}_i$ to force them to lie on the $d$-dimensional sphere $\sqrt{d}\,\mathbb{S}^{d-1}$. More precisely, $\hat{x}_i \leftarrow \sqrt{d} \cdot \hat{x}_i/\|\hat{x}_i\|_2$, thus respecting the fact that minimization should happen on the sphere as in (164). Unless stated otherwise, the step size is set by default to $2 \cdot 10^3$ for the CIFAR-10 experiments and to 20 for the experiments on synthetic data. On Tiny-ImageNet we use a step size of $8 \cdot 10^3$. Momentum is set to $0.9$. We consider the reconstruction optimization converged when the normalized reconstruction loss $\mathcal{L}(\hat{X})/\|\theta^*\|_2^2$ drops below $10^{-7}$. Normalizing by $\|\theta^*\|_2^2$ makes the loss of order 1 at the beginning of the optimization so that the chosen threshold of $10^{-7}$ corresponds to the effective numerical resolution in floating point representation. Unless explicitly mentioned otherwise, in every experiment we perform, the optimization is run until the reconstruction loss has converged.

**Statistical robustness.** To ensure that our findings are not tied to a particular random initialization, all experiments are repeated over multiple random seeds. Each seed induces independent initializations of network parameters and of the reconstruction variables $\hat{X}$. All the quantitative results reported in the paper correspond to averages (and variability) across these runs.

**CIFAR-10 protocol.** For random features experiments, we construct a balanced subset from the CIFAR-10 (Krizhevsky et al., 2009) training split by iterating once through the data and collecting the

first $n/2$ occurrences of the "*frog*" class (negative) and $n/2$ of the "*truck*" class (positive), yielding $n$ samples. Each image is flattened to $d = 3 \cdot 32 \cdot 32 = 3072$ and concatenated into $X \in \mathbb{R}^{n \times d}$. We standardize using subset statistics computed over the selected $n$ examples. Targets are $Y \in \{\pm 1\}^n$, with the first $n/2$ entries set to $-1$ and the remaining $n/2$ to $+1$. We use "*frog*" vs. "*truck*" purely as a convenient benchmark. Our conclusions do not hinge on category semantics, and we observed the same qualitative behavior across alternative class pairs. For multi-class experiments, we use the same balanced construction; targets are one-hot encoded as 10-dimensional vectors with a single 1 at the true class index (and 0 elsewhere). Also for ResNet and vision transformer runs we use the same protocol as for random features, but images are kept in tensor form ($\mathbb{R}^{3 \times 32 \times 32}$) and normalized per channel using the full CIFAR-10 training split statistics (mean $[0.4914, 0.4822, 0.4465]$, std $[0.247, 0.243, 0.261]$).

**Tiny-ImageNet protocol.** We construct a balanced subset from the Tiny-ImageNet (Le & Yang, 2015) training split by iterating once through the data and collecting the first $n/2$ occurrences of *class 2* (negative) and $n/2$ of *class 116* (positive), yielding $n$ samples. Each image is flattened to $d = 3 \cdot 64 \cdot 64 = 12288$ and concatenated into $X \in \mathbb{R}^{n \times d}$. We standardize using subset statistics computed over the selected $n$ samples. Targets are $Y \in \{\pm 1\}^n$, with the first $n/2$ entries set to $-1$ and the remaining $n/2$ to $+1$. We have randomly drawn *class 2* vs. *class 116* and used this pair purely as a convenient benchmark. Also in this case, our conclusions do not hinge on category semantics.

### D.1 ADDITIONAL DETAILS ON DEEP RESIDUAL NETWORKS

In this section, we give precise implementation details of the class of deep residual networks we consider. We focus on ResNet-style architectures (He et al., 2016) adapted to CIFAR-10 images while preserving the canonical residual topology. Specifically, we remove batch normalization and max pooling layers so that feature map resolution and statistics are maintained throughout. Apart from this, the residual block structure and overall layout are preserved. More formally, let $\phi(\cdot) = \mathrm{ReLU}(\cdot)$ applied element-wise and let $C_{a \to b}^{k \times k}[\cdot]$ denote a bias-free 2D convolution with kernel size $k \times k$, stride 1 and padding 1, mapping from $a$ to $b$ channels. Let a residual block be

$$B(u) = \phi(u + C_{h \to h}^{3 \times 3}[\phi(C_{h \to h}^{3 \times 3}[u])]) . \tag{165}$$

Given an image $x \in \mathbb{R}^{3 \times 32 \times 32}$ (as in the case of CIFAR-10 examples), we consider in our experiments deep residual networks with architecture

$$z_0 = C_{3 \to h}^{28 \times 28}[x] \qquad z_b = B(z_{b-1}) \ \text{ for } b = 1, \dots, 4 , \qquad f_{\mathrm{RN}}(x) = [\theta^{(L)}]^\top \mathrm{vec}(z_4) , \tag{166}$$

with $\theta^{(L)} \in \mathbb{R}^{h \cdot 7 \cdot 7}$ the last layer's parameters. We denote with $\mathrm{vec}(\cdot)$ the flattening of the dimensions of the 3D tensor $z_4$ to a vector. The choice of dimensionality for $\theta^{(L)}$ is motivated by the first convolution mapping the spatial size for CIFAR-10 images to $h \times 7 \times 7$ and the blocks preserving it. We select the number of output channels $h$ based on the choice of the number of parameters in the last layer $p^{(L)}$ as these two quantities are tied together, being $h = p^{(L)}/7^2$, eventually truncated to the nearest integer. Parameters of the first convolutional layer are initialized as $\theta_{i,j}^{(0)} \sim \mathcal{N}(0, 1/d)$ i.i.d., while for $l > 1$ parameters are initialized as $\theta_{i,j}^{(l)} \sim \mathcal{N}(0, 1/\mathrm{fan\_in})$ i.i.d. Here, $\mathrm{fan\_in}$ equals the number of input units (for convolutional layers: $h \cdot k^2$; for the last linear layer: $p^{(L)}$).

## E ADDITIONAL NUMERICAL RESULTS

### E.1 FURTHER RESULTS ON SPAN INCLUSION

As mentioned at the end of Section 4 and in the caption of Figure 2, we test whether the features computed on the training dataset are spanned by the features of the reconstructed examples. To this end, we calculate the average per-feature orthogonal residual, formally defined as

$$r(\hat{\Phi}) = \frac{1}{n\sqrt{p}} \sum_{i=1}^{n} \left\| P_{\hat{\Phi}}^{\perp} \, \varphi(x_i) \right\|_2 , \tag{167}$$

where $P_{\hat{\Phi}}^{\perp} = I - P_{\hat{\Phi}}$ projects onto the orthogonal complement of $\mathrm{span}\{\mathrm{rows}(\hat{\Phi})\}$. Thus $r(\hat{\Phi}) = 0$ if and only if every $\varphi(x_i)$ lies in $\mathrm{span}\{\mathrm{rows}(\hat{\Phi})\}$. The normalization by $n\sqrt{p}$ makes $r(\hat{\Phi})$ of order 1 so that values of $r(\hat{\Phi}) \ll 1$ indicate numerically negligible residuals (*i.e.*, effective span inclusion).

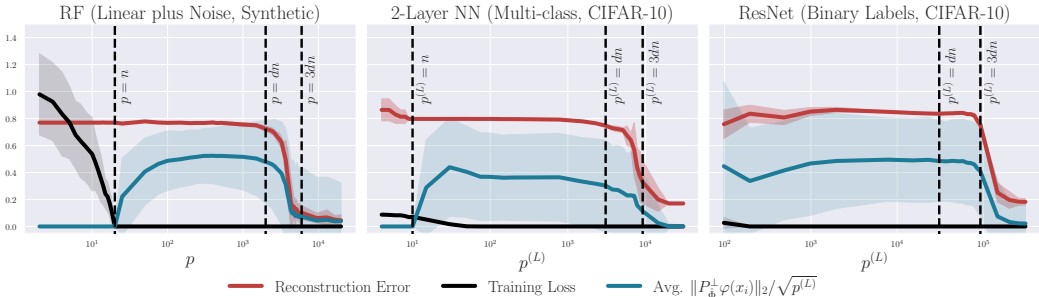

Figure 9: **Features of the training dataset $\Phi$ are spanned by the features of the reconstructed dataset $\hat{\Phi}$.**
We consider RF regression with ReLU activation on i.i.d. samples uniformly distributed on the $d$-dimensional
sphere ($d = 100, n = 20$), 10-class one-hot regression with a 2-layer ReLU network trained with gradient
descent on CIFAR-10 ($n = 10$) and regression on binary labels (*frogs vs. trucks*) with a ResNet trained with
gradient descent on CIFAR-10 ($n = 10$). We report mean (solid line) and standard deviation (shaded area) for
the reconstruction error (in red), for the training loss (in black) and for the per-feature orthogonal residual of
projecting $\Phi$ onto $\mathrm{span}\{\mathrm{rows}(\hat{\Phi})\}$ (in blue), as the number of parameters increases. We indicate with $p^{(L)}$ the
number of parameters in the last layer. Statistics are computed across 10 distinct random seeds.

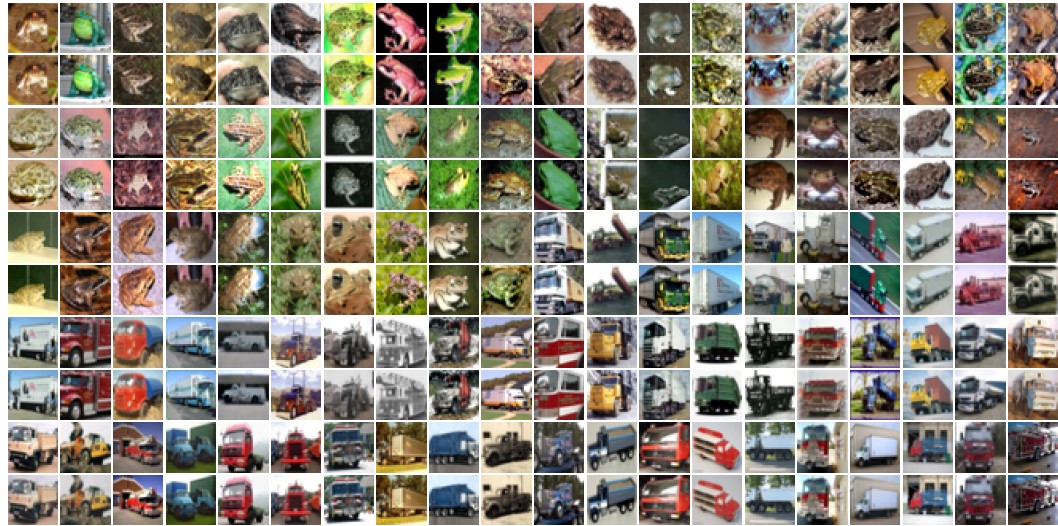

Figure 10: **Reconstructing CIFAR-10 training images from an RF model with the activation function
satisfying Assumption 2.** We repeat Figure 4 with the same seed, reconstructing CIFAR-10 training images
($n = 100$) from an RF model with activation function $\phi(z) = \mathrm{ReLU}(z) + \tanh(z)$ on binary labels at
$p = 10dn$. Odd rows report the ground truth images, while even rows the reconstructed ones. In this experiment
the activation function is consistent with Assumption 2, and indeed, this excludes the possibility of reconstructing
sign-flipped versions of the originals.

Figure 9 summarizes the interpolation-reconstruction behavior across RF regression (synthetic data,
same setting of Figure 3), two-layer ReLU neural networks (CIFAR-10, multi-class, same setting
of left Figure 6), and deep residual networks (CIFAR-10, binary, same setting of right Figure 6).
Both the reconstruction error (red) and the per-feature orthogonal residual (blue) remain large until
the model crosses the threshold $p \approx dn$ ($p^{(L)} \approx dn$ for neural networks), after which they drop
sharply. This trend is consistent with Figure 2: successful reconstruction occurs precisely when
$\varphi(x_i) \in \mathrm{span}\{\mathrm{rows}(\hat{\Phi})\}$ for all $i$, *i.e.*, when $\|P_{\hat{\Phi}}^{\perp}\varphi(x_i)\|_2 \approx 0$. Also in these settings, for $p \leq n$, the
optimization converges to the non-degenerate case $P_{\hat{\Phi}} = I$, which makes the orthogonal residual
trivially zero.

### E.2 DATA RECONSTRUCTION ON RF WITH MIXED-PARITY ACTIVATIONS

Figure 10 repeats the setup of Figure 4 on CIFAR-10 (binary labels, $n = 100$) using the *same random
seed* and optimization protocol, but replacing ReLU with the activation $\phi(z) = \mathrm{ReLU}(z) + \tanh(z)$

that has high-order Hermite coefficients of different parities. In contrast to Figure 4, no sign-flipped reconstructions appear. This outcome is precisely what Assumption 2 and Remark 1 predict: ReLU alone violates the assumption (its odd Hermite coefficients $\mu_{2\ell+1}$ vanish for $\ell > 1$), allowing $x_i$ and $-x_i$ to be indistinguishable under the reconstruction loss. Adding $\tanh$ supplies non-zero coefficients of opposite parity, restoring identifiability of the sign and removing this degeneracy.

### E.3 Additional ablation studies

**Effects of different learning rates in the reconstruction optimization.** To assess the effect of the step size on the success of the data reconstruction, we have conducted an additional ablation using Binary CIFAR-10 ($n = 20$) on the RF model. Starting from the base step size $\eta^* = 2 \cdot 10^3$, we have considered $\eta \in \{\eta^*/4, \eta^*/2, \eta^*, 2\eta^*, 4\eta^*\}$ across 10 distinct random seeds. In Figure 11, we plot the reconstruction error, the total number of iterations for which the reconstruction algorithm is run, and the final reconstruction loss $\mathcal{L}$, as a function of the number of parameters $p$. For larger step sizes ($2\eta^*$ and $4\eta^*$), we observe oscillatory behavior of the reconstruction loss that prevents convergence within a reasonable budget. Guided by the smallest step size ($\eta^*/4$), for which the loss consistently converges to zero within at most $6 \cdot 10^4$ iterations across seeds and number of parameters $p$, we cap the optimization at $6 \cdot 10^4$ iterations in this experiment. As shown in Figure 11, for all step sizes that converge within this budget ($\eta^*/4, \eta^*/2, \eta^*$), the reconstruction error drops to zero once $p \gg dn$, indicating that the reconstruction procedure is robust to the choice of the step size provided it is not taken excessively large.

**Number of reconstructed samples different from number of training samples.** So far, we only discussed the results of optimizing $\mathcal{L}(\hat{X})$ when setting $\hat{X}$ to be a matrix in $\mathbb{R}^{n \times d}$. In Figure 12, we numerically investigate the effects of setting $\hat{X}$ to be a matrix of size $\hat{n} \times d$, when $\hat{n} \neq n$. Specifically, we consider Binary CIFAR-10 with $n = 50$ (25 "frog" images vs. 25 "truck" images) and a RF model with $p = 10dn$. When reconstructing fewer samples ($\hat{n} < n$), we observe that the outputs of the reconstruction often mix the structure coming from multiple ground-truth images. Intuitively, this translates in the fact that the reconstruction seems to be minimized when $\hat{X}$ has rows that are a superposition of multiple training data, rather than a subset of the training data themselves. Increasing $\hat{n}$ progressively reduces this noise. Conversely, when reconstructing with more rows than training samples ($\hat{n} > n$), we find that 50 of the recovered images match the $n = 50$ training samples, while the extra $\hat{n} - n = 10$ reconstructions are simply duplicates. From a practical perspective, this means that, in order to estimate $n$, it suffices to iteratively increase $\hat{n}$ until the first duplicate appears.

**Reconstruction from vision transformer architecture.** In Figure 13, we provide numerical results on vision transformers trained on CIFAR-10 in the multi-class setting, using the same reconstruction procedure. We study randomly initialized vision transformers (Dosovitskiy et al., 2021) akin to ViT-B/16 on which we vary the embedding dimension trained on one-hot encoded labels from the first five classes of the CIFAR-10 dataset ($n = 5$). Vision transformers display a similar phenomenology to fully connected and residual architectures: when the number of parameters $p^{(L)}$ exceeds $dn$, the reconstruction error goes down and images are recovered successfully.

**Effects of weight regularization.** So far, we discussed reconstruction from the weights of a trained model without regularization, as expressed in Eq. (2). Adding a ridge parameter $\lambda > 0$ would change the training loss as

$$\mathcal{L}_{train}(\theta) = \frac{1}{n} \sum_{i=1}^{n} \left( \varphi(x_i)^\top \theta - y_i \right)^2 + \lambda \|\theta\|_2^2, \tag{168}$$

whose unique minimizer is

$$\theta^* = \Phi^\top \left( \Phi\Phi^\top + n\lambda I \right)^{-1} Y. \tag{169}$$

For a fixed value of $\lambda$ (not dependent on $d, n$ and $p$), and due to Eq. (14), we have

$$\|n\lambda I\|_{\text{op}} = O(n), \qquad \lambda_{\min}\left( \Phi\Phi^\top \right) = \Omega(p). \tag{170}$$

Intuitively this suggests that, as $p$ grows large, the effect of a fixed regularization term $\lambda$ becomes negligible, and reconstruction is still possible. This is verified numerically in Figure 15, where we see that even for very large values of $\lambda$ we find a qualitatively similar threshold for successful reconstruction.

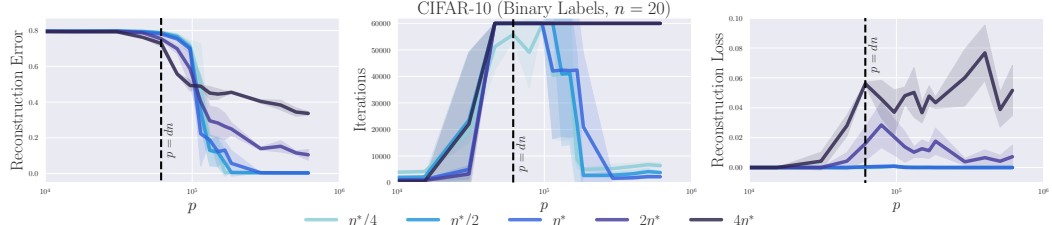

Figure 11: **Ablation on step size used for the reconstruction procedure.** We consider RF regression with ReLU activation on binary labels (*frogs vs. trucks*) with $n = 20$ images (10 examples per class) from CIFAR-10. Starting from the base step size $\eta^* = 2 \cdot 10^3$ used in all the CIFAR-10 experiments, we report the reconstruction error (left), total number of training iterations needed by the reconstruction algorithm (center) and reconstruction loss (right) as a function of the number of parameters $p$, aggregated over 10 distinct random seeds. Guided by the smallest step size ($\eta^*/4$), for which the loss consistently converges to zero within at most $6 \cdot 10^4$ iterations across seeds and number of parameters $p$, we cap the optimization at $6 \cdot 10^4$ iterations.

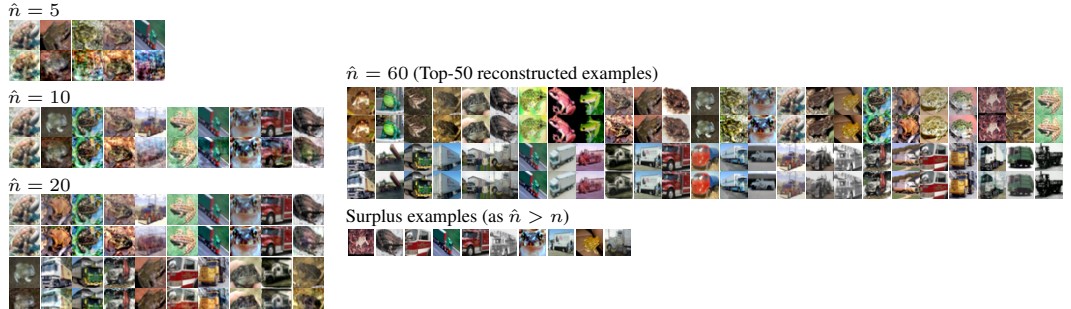

Figure 12: **Dataset reconstruction without knowing the exact number of training samples $n$.** We fit an RF model with ReLU activation and $p = 10dn$ on binary labels (*frogs vs. trucks*) with $n = 50$ images (25 examples per class) from CIFAR-10. We then optimize the reconstruction loss $\mathcal{L}(\hat{X})$ with $\hat{X} \in \mathbb{R}^{\hat{n} \times d}$ and the number of reconstructed samples $\hat{n}$ different from $n$. We assess both the case when reconstructing fewer samples ($\hat{n} < n$, left column) and more samples ($\hat{n} > n$, right column) than the ground-truth number of samples $n$. In the latter case, we also plot the extra $\hat{n} - n = 10$ examples.

**Effects of pruning.** In Figure 16, we explore numerically the behavior of our reconstruction algorithm on a pruned network, following the same synthetic setup of Figure 3. We first obtain $\theta^* = \Phi^+ Y$ and then run Optimal Brain Surgeon (Hassibi & Stork, 1992), which in our context (convex problem on square loss) provably incurs in the minimal increase in training loss, given a fixed sparsity ratio. We select either $d = 30, n = 20$ (left plot of Figure 3) or $d = 100, n = 100$ (right plot of Figure 3), reporting the results for several sparsity ratios. As expected, pruning seems to increase the number of parameters needed for data reconstruction.

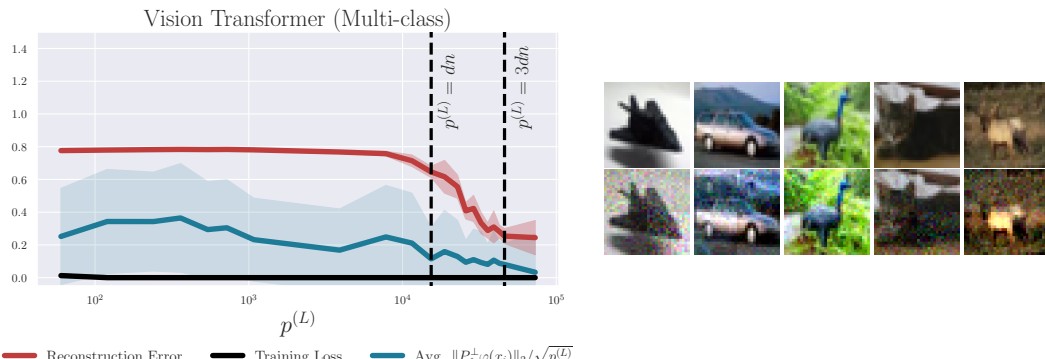

Figure 13: **Thresholds for label fitting and data reconstruction for Vision Transformers trained with gradient descent on** $n = 5$ **CIFAR-10 images.** (Left) We consider regression with the square loss, training Vision Transformers on the one-hot encoding of 5-class labels. We report mean (solid line) and standard deviation (shaded area) for the reconstruction error (in red), for the training loss (in black) and for the per-feature orthogonal residual of projecting $\Phi$ onto $\mathrm{span}\{\mathrm{rows}(\hat{\Phi})\}$ (in blue), as the number of parameters in the last layer $p^{(L)}$ increases. Statistics are computed across 10 distinct random seeds. (Right) Results of the reconstruction when $p^{(L)} = 4dn$. The first row reports the ground truth images, while the second row the reconstructed ones.

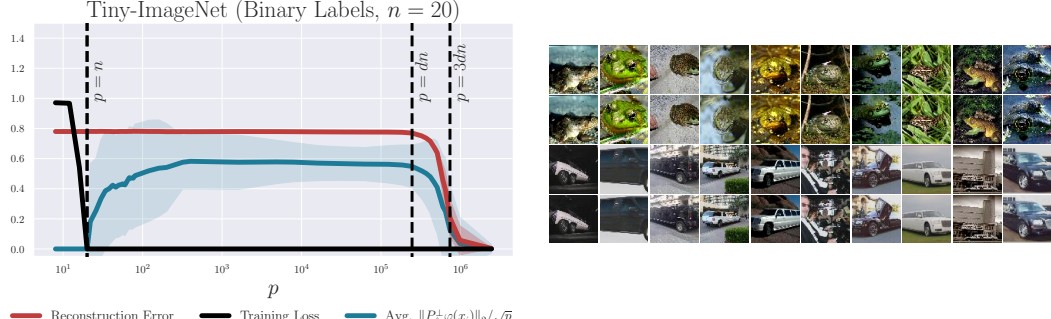

Figure 14: **Thresholds for label fitting and data reconstruction on Tiny-ImageNet.** (Left) We consider RF regression with ReLU activation on binary labels (*class 2 vs. class 116*) with $n = 20$ images (10 examples per class) from Tiny-ImageNet ($d = 12288$). We report mean (solid line) and standard deviation (shaded area) for the reconstruction error (in red), for the training loss (in black) and for the per-feature orthogonal residual of projecting $\Phi$ onto $\mathrm{span}\{\mathrm{rows}(\hat{\Phi})\}$ (in blue), as the number of parameters $p$ increases. Statistics are computed across 10 distinct random seeds. (Right) Results of the reconstruction when $p = 10dn$. Odd rows report the ground truth images, while even rows the reconstructed ones which are all visually very similar.

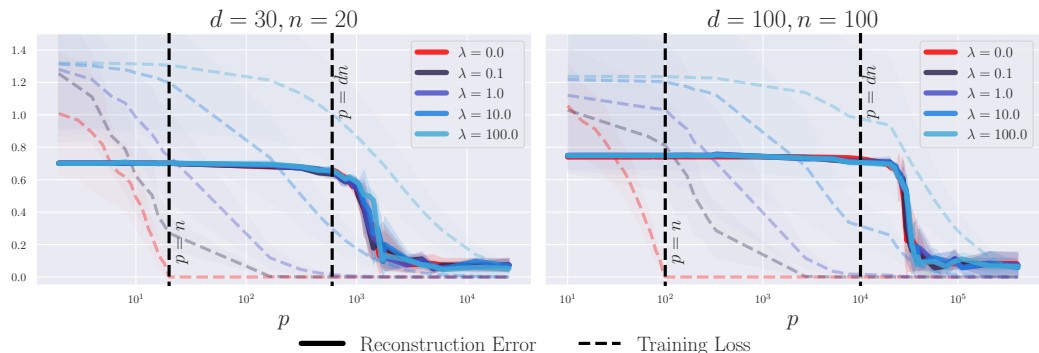

Figure 15: **Thresholds for label fitting and data reconstruction in the presence of regularization.** We consider RF regression with ReLU activation, fitting a noisy linear model with $\ell_2$ penalty (ridge). The training data is i.i.d. uniformly drawn from the $d$-dimensional sphere. We report mean (solid/dashed lines) and standard deviation (shaded areas) for both the reconstruction error (solid lines) and training loss (dashed lines) as the number of parameters $p$ increases, at different choices of input dimensions $d$ and number of samples $n$. Statistics are computed across 10 distinct random seeds. Lighter colors refer to stronger regularization, which does not significantly deviate from the behavior of training without regularization (red lines, occluded).

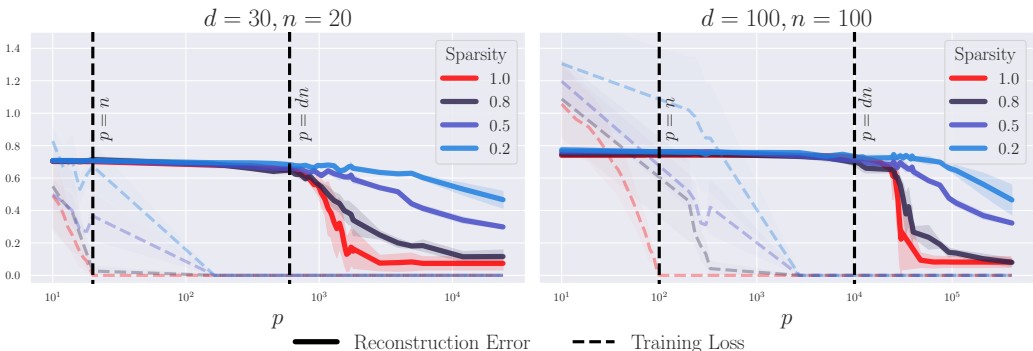

Figure 16: **Thresholds for label fitting and data reconstruction of pruned models.** We consider RF regression with ReLU activation, fitting a noisy linear model. The training data is i.i.d. uniformly drawn from the $d$-dimensional sphere. After fitting, we prune the trained parameters $\theta^*$ with Optimal Brain Surgeon (Hassibi & Stork, 1992) to a desired sparsity ratio (*i.e.*, $p'/p$ with $p'$ the number of remaining parameters). We report mean (solid/dashed lines) and standard deviation (shaded areas) for both the reconstruction error (solid lines) and training loss (dashed lines) as the number of parameter $p$ increases, at different choices of input dimensions $d$ and number of samples $n$. Statistics are computed across 10 distinct random seeds. Lighter colors refer to lower sparsity ratios, compared against the unpruned baselines (red lines).

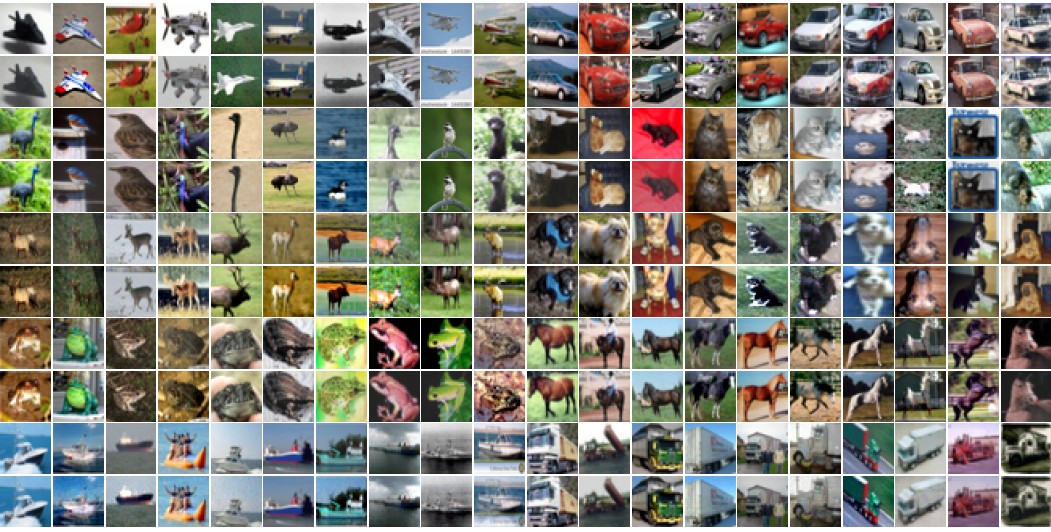

Figure 17: **Multi-class training data reconstructed from a neural network trained with cross-entropy on CIFAR-10.** We repeat the experiment of Figure 7 by training a two-layer ReLU network on $n = 100$ images from CIFAR-10 dataset (10 examples per class) with gradient descent. In this experiment, we cast the training procedure as multi-class classification on *cross-entropy loss*. The number of parameters in the last layer of the network is $p^{(L)} = 4dn$.

