# OpenReview forum: "A Law of Data Reconstruction for Random Features (And Beyond)"
_ICLR.cc/2026/Conference — ICLR 2026 Poster_

### Official Review · Reviewer_hVw8 · 2025-10-21

**Soundness:** 4
**Presentation:** 3
**Contribution:** 3
**Rating:** 6
**Confidence:** 4

**Summary:**

The paper considers a random-features model, and proves that if the number of parameters $p$ of the output layer is larger than the total number of coordinates in the training dataset ($nd$), then the linear subspace spanned by the features of the training examples gives enough information for (approximately) reconstructing training examples. This requires certain assumptions, including some assumptions on the Hermite expansion of the activation function. Inspired by their theoretical results, they suggest a method for reconstruction without knowing the subspace of training features. They demonstrate how their method works in various settings. Also, they show how empirically p=n is the threshold for interpolation while p=nd is the threshold for reconstructability.

**Strengths:**

The theoretical results (Theorems 1 and 2) are very nice and non-trivial. I think that understanding the theoretical aspects of the ability to reconstruct training data is a challenging and intriguing question, and the paper makes a step in this direction.

Regarding the empirical results, I liked the consistent thresholds of p=n for interpolation vs. p=nd for reconstruction. The authors show that these thresholds appear in various settings, which improves our understanding of this aspect.

**Weaknesses:**

My main concern is that the implications of the results on reconstruction in the common setting, where the attacker does not know the subspace of training features, are somewhat limited for the following reasons:

First, the theoretical results only consider the case where the attacker knows this subspace, and hence it is unclear whether the attacker can reconstruct training data without this knowledge. The theoretical results imply that it suffices to know the subspace of training features, which is interesting and non-trivial, but it does not allow for reconstruction only from the network’s parameters, as in “common” reconstruction attacks.

Regarding the reconstruction algorithm that does not assume knowledge of the training features subspace: The proposed method relies on the fact that the output weights are spanned by the features. This observation is similar to Loo et al. (2024), as the authors noted. Thus, this method does not rely on the theoretical results, and it could be argued that they are independent of the theoretical part of the paper. The fact that it is similar to Loo et al. also makes it less novel (although the setting here is different). Having said that, I still think that the empirical results here give convincing evidence regarding the importance of the p=nd threshold. This was not studied in such a systematic way in previous works, and I view it as a valuable observation.

Overall, I think that the theoretical results are intriguing and nontrivial, and that the conceptual contribution regarding the “law of data reconstruction”, which appears both in the theoretical results and in the experiments, is valuable. But given the limitations discussed above, I tend to “weak accept”.

A more minor comment: In the paper, the authors mention that for interpolation, we need more parameters than training samples. A more accurate claim is that the number of parameters times the depth needs to be larger than the number of samples. Or alternatively, that the number of bits in the network’s representation needs to be larger than the number of samples. See the paper “On the Optimal Memorization Power of ReLU Neural Networks” by Vardi, Yehudai, and Shamir for details.

**Questions:**

The paper only considers the square loss function. However, the empirical part may also work for classification with the logistic or cross-entropy loss. That is, Eq. 11 should still hold for classification (since the max margin predictor is spanned by the inputs). Have the authors verified whether the “law of data reconstruction” also appears in classification?

---

> ### Author Response · Authors · 2025-11-20
>
> We thank the reviewer for the positive evaluation of our work. We address the question and the more minor comment below. We have edited our paper accordingly and uploaded a revised version, where the changes are marked in red color to make them more visible.
>
> ---
>
> **More minor comment**:  _Parameters needed for interpolation_
>
> Good point, we have added the following in l. 111-114 of the revision: “Under a separability assumption, it has also been shown that $n$ arbitrary training points can be perfectly classified by a neural network with $p \gg \sqrt{n}$ parameters (Vardi et al., 2022). This, in the context of regression, implies that $p \gg \log(1 / \epsilon) \sqrt{n}$ is sufficient to guarantee an error of at most $\epsilon$.”
>
> ---
>
> **Question**: _Regression vs Classification_
>
> This is a great question. We run additional numerical simulations both on logistic regression for synthetic data and multiclass classification (CIFAR-10) training a 2-layer neural network with cross entropy loss. The results can be found in Figure 8 (in the main body of the revision) and Figure 17 (in Appendix E.3 of the revision), and they provide evidence that the law of data reconstruction also appears in classification.
>
> More precisely, we first consider the synthetic task where the training data $x_i$ are $n=100$ i.i.d. samples drawn uniformly on the sphere of radius $\sqrt{d}=10$. The labels are given by $y_i= \text{sign}(g^\top x_i)$, where $g\in\mathbb{R}^d$ has entries $g_i \sim \mathcal{N}(0,1/d)$. We compute $\theta^*$ minimizing the logistic loss $\ell(\theta):=\sum_{i=1}^n \log(1+e^{-y_i \varphi(x_i)^\top \theta})$ with gradient descent, and consider the same reconstruction algorithm as in Eq. (11). In the left panel of Figure 8, we find the usual threshold $p \approx dn$ for successful reconstruction of the training set.
>
> In the right panel of Figure 8, we consider a two-layer neural network trained with cross-entropy loss on $n = 10$ samples from the 10 classes of CIFAR-10. We see that the reconstruction algorithm yields similar results as the ones for regression in Figure 3.
>
> We have incorporated this discussion in l. 483-510 of the revision.

---

> > ### Comment · Reviewer_hVw8 · 2025-11-20
> >
> > Thank you for your response. I will maintain my evaluation.

---

### Official Review · Reviewer_Krj1 · 2025-10-27

**Soundness:** 3
**Presentation:** 2
**Contribution:** 3
**Rating:** 6
**Confidence:** 3

**Summary:**

This paper proposed a new law of data reconstruction in overparameterized neural networks. The authors point out that when model parameters contain sufficient information to reconstruct the entire training set. The key finding is that full data reconstruction becomes feasible when the number of parameters 𝑝 greatly exceeds the product of data dimension 𝑑 and sample count 𝑛.

**Strengths:**

1. Clear statement of the reconstruction law with 2 new theorem.
2. Proposed the first quantitative relationship between model size and data reconstructability.
3. Visualization validation from light-weight models convincingly support the claims.

**Weaknesses:**

1. Analysis restricted to random features and small networks (even no evidence from the widely-used transformer to show whether it also works on the attention framework), not large-scale modern models.
2. Experiments involve small sample sizes (mainly on 32*32), limiting statistical robustness on larger size image from STL10 or Imagenet.

**Questions:**

1. Does this theorym could transfer to transformer or diffusion models?
2. Will this thoerm still work when the training datapoints become morem complex with larger size?
3. Will regularization or pruning invalidate the reconstruction law?

---

> ### Author Response · Authors · 2025-11-20
>
> We thank the reviewer for the positive evaluation of our work. We address the three questions below. We have edited our paper accordingly and uploaded a revised version, where the changes are marked in red color to make them more visible.
>
> ---
>
> **Question 1**: _Transferability of the results to transformer or diffusion models_
>
> Thank you for this observation. To address it and broaden the empirical scope of our work, we give additional numerical results on vision transformers trained on CIFAR-10 in the multi-class setting, using the same reconstruction procedure. The results are reported in Fig. 13 of the revision, which is discussed in l. 1922-1927.
>
> We study randomly initialized Vision Transformers (Dosovitskiy et al., 2021), akin to ViT-B/16 on which we vary the embedding dimension, trained on one-hot encoded labels from the first five classes of the CIFAR-10 dataset ($n=5$). The models are optimized using gradient descent to minimize the squared loss. We report thresholds for reconstruction as we increase the number of parameters in the final layer $p^{(L)}$, alongside a qualitative result when $p^{(L)} = 4dn$.
>
> At a high level, vision transformers display a similar phenomenology to fully connected and residual architectures: when the number of parameters $p^{(L)}$ exceeds $dn$, the reconstruction error goes down and images are recovered successfully.
>
> ---
>
> **Question 2**: _Experiments are mainly on sizes 32 * 32, will the theorems still work when the training datapoints become larger?_
>
> Thank you for the question. First, we note that our theoretical results hold for _arbitrarily large_ values of $d$, as long as $p \gg dn$.
>
> Second, while our experiments focus on inputs with dimensionality $3 \times 32 \times 32 = 3072$ for computational convenience, we do expect similar results to hold for datasets having a larger dimensionality. In the revision, we complement the observations reported in Figure 1 by conducting new experiments on Tiny-ImageNet, which has input dimension of order $10^4$. We fit random features models on binary classes ($n=20$, 10 examples from class 2 and 10 from class 116; the classes are an arbitrary choice, as we don’t hinge on semantics). We report in the left part of Fig. 14 the final reconstruction error, training loss and average normalized per-feature orthogonal residual at increasing values of $p$, aggregated across four distinct random seeds. Also in this scenario, there is a transition around $p=dn$ after which both the reconstruction error and the average per-feature orthogonal residual vanish. The right part Fig. 14 then shows reconstruction examples when $p=10dn$, which are perceptually indistinguishable from original training data.
>
> We have edited the revision at l. 417-419 to incorporate this discussion, and displayed the reconstructed images also in Figure 5 of the body.
>
> ---
>
> **Question 3**: _Will regularization or pruning invalidate the reconstruction law?_
>
> Thank you for the interesting question. Consider minimizing the training square loss of a regularized random features model
>
> $$\mathcal L_{train}(\theta) = \frac{1}{n} \sum_{i = 1}^n \left( \varphi(x_i)^\top \theta - y_i \right)^2 \\, + \\, \lambda \|| \theta \||_2^2.$$
>
> This yields the unique solution
>
> $$\theta^* = \Phi^\top \left( \Phi \Phi^\top + n \lambda I  \right)^{-1} Y,$$
>
> which matches Equation (2) if we set $\lambda = 0$. Then, for a fixed value of $\lambda$ (not dependent on $d, n$ and $p$), and due to Eq. (14), we have
>
> $$\lambda_{\rm max} ( n \lambda \\, I  ) = O(n),  \qquad   \lambda_{\rm min}(\Phi \Phi^\top) = \Omega(p).$$
>
> Intuitively this suggests that, as $p$ grows large, the effects of a fixed regularization term $\lambda$ becomes negligible, and reconstruction is still possible. This is verified numerically in Figure 15, where we see that even for very large values of $\lambda$ we find a qualitatively similar threshold for successful reconstruction. We included this discussion in l. 1929-1943, and mentioned this result in l. 515.
>
>
>
>
>
>
> In contrast, pruning seems to increase the number of parameters needed for data reconstruction: we explore numerically this behavior in a synthetic setup of Figure 3. For pruning we first obtain $\theta^* = \Phi^+ Y$ and then run Optimal Brain Surgeon (Hassibi & Stork, 1992), which in our context (convex problem on square loss) provably incurs in the minimal increase in training loss, given a fixed sparsity ratio. We select either $d=30, n=20$ (left plot of Figure 16) or $d=100, n=100$ (right plot of Figure 16), reporting the results for several sparsity ratios. We included this discussion in l. 1977-1983, and mentioned this result in l. 517.

---

> > ### Comment · Reviewer_Krj1 · 2025-11-20
> >
> > Thank you very much for your rebuttal. Your new experiment results are impressive. I am more confident about the quality of this paper now, I will raise my score and confidence for this paper.

---

### Official Review · Reviewer_heHJ · 2025-10-30

**Soundness:** 3
**Presentation:** 4
**Contribution:** 3
**Rating:** 8
**Confidence:** 3

**Summary:**

The paper studies data memorization as the ability to reconstruct training inputs from model parameters. It shows that, unlike label memorization (which occurs when $p \gg n$), full input reconstruction requires $p \gg nd$, where $p$ is the number of parameters, $n$ the number of samples, and $d$ the data dimension. The authors prove that in random features models, if the span of the features of $X \in \mathbb{R}^{n\times d}$ lies inside the span of the features of $\hat{X}$, then (up to permutation) $\hat{X}$ must match $X$ with high probability. They also propose a reconstruction loss and show empirically that the same scaling enables recovery in random features, two-layer networks, and deep residual networks.

**Strengths:**

1. The paper is clearly structured and well written, and it is easy to follow.
2. Data memorization, in the sense of recovering training inputs from a trained model, is an important problem with implications for privacy and model safety. The paper addresses it in a timely way.
3. The theoretical and empirical results are convincing. Although the theory is developed for random feature models with assumptions on the nonlinearity, it is compelling that similar empirical behavior appears in more complex architectures under the same scaling.
4. To my knowledge, both the analysis and the reconstruction algorithm motivated by the theory are novel, and the results are convincing.

**Weaknesses:**

1. The reconstruction algorithm, although motivated by the theory, still feels somewhat mysterious. In particular, the trained $\theta*$ is only known to be a vector in the span of the features of the training data. Enforcing that $\theta*$ lies in the span of the reconstructed features does not immediately imply that $\varphi(x_i)$ is in $\text{span}(\varphi(\hat{x}_j))$ for all $i$. It would be helpful if the authors could clarify this.
2. The theory appears to require knowing the exact number of training samples $n$. Is this correct? What would happen if $\hat{X}$ had a different number of rows than $X$?
3. In the proof sketch of Theorem 1 (lines 227–229), it seems that the roles of $\hat{x}$ and $x$ might be swapped. Can the authors clarify whether this is a typo?

**Questions:**

Please refer to the previous section

---

> ### Author Response · Authors · 2025-11-20
>
> We thank the reviewer for the positive evaluation of our work. We address the three weaknesses below. We have edited our paper accordingly and uploaded a revised version, where the changes are marked in red color to make them more visible.
>
> ---
>
> **Weakness 1**: _Enforcing that $\theta^*$ lies in the span of the reconstructed features does not immediately imply that $\varphi(x_i)$ is in span$(\varphi(\hat x_i))$ for all $i$_
>
> We agree with the reviewer that enforcing that $\theta^\*$ lies in the span of the reconstructed features does not immediately imply that $\varphi(x_i)$ is in span$( \\{ \varphi(\hat x_j) \\}_{j=1}^n )$ for all $i$. However, numerical experiments support that enforcing $\theta^\*$ being on the span of the reconstructed features is _sufficient_ to have that $\varphi(x_i)$ is in the span of $\varphi(\hat x_j))$ for all $i$, see the blue line approaching 0 for $p \gg dn$ in Figure 2. Additional numerical evidence pointing in the same direction is reported in Figure 9 in the appendix. We also conjecture that this _sufficiency_ can be shown theoretically, and in l. 525-530 in the Conclusions we cast this as a future research direction. We have further clarified this point in l. 317-347 of the revision.
>
> ---
>
> **Weakness 2**: _The theory appears to require knowing the exact number of training samples $n$_
>
> Thank you for raising this interesting point. Denoting with $\hat n$ the number of rows of $\hat X$, we numerically studied the behavior resulting from optimizing the reconstruction loss when $\hat n \neq n$. Specifically, we have run additional experiments on Binary CIFAR-10 fixing $n=50$ (25 "frog" images vs. 25 "truck" images) and considering a random features model with $p=10dn$.
>
> As shown in Fig. 12, when reconstructing fewer samples ($\hat n < n$), we observe that the outputs of the reconstruction often mix the structure coming from multiple ground-truth images. Intuitively, this translates in the fact that the reconstruction seems to be minimized when $\hat X$ has rows that are a superposition of multiple training data, rather than a subset of the training data themselves. Increasing $\hat n$ progressively reduces this noise.
>
> Conversely, when reconstructing with more rows than training samples ($\hat n > n$), we find that $50$ of the recovered images match the $n=50$ training samples, while the extra $\hat n - n=10$ reconstructions are simply duplicates. From a practical perspective, this means that, in order to estimate $n$, it suffices to iteratively increase $\hat n$ until the first duplicate appears.
>
> We discuss this in Appendix E.3 of the revision, and mention this behavior at l. 512-514.
>
> ---
>
> **Weakness 3**: _Roles of $x$ and $\hat x$ in the proof sketch of Theorem 1_
>
> Thank you for your question. No, this is not a typo. For $p \gg n$, we know that the smallest eigenvalue $\lambda_{\min}(\Phi \Phi^\top) > 0$. Thus, the rows of $\Phi$ are linearly independent (they span a sub-space of dimension _exactly_ $n$). Then, if the $n$ row vectors of $\hat \Phi$ indeed span all rows of $\Phi$ (as by the hypothesis of Theorem 1), it must hold that ${\rm span}\\{{\rm rows}(\Phi)\\} = {\rm span}\\{{\rm rows}(\hat \Phi)\\}$, which allows us to write Equation (5). We have clarified this in l. 228-229 of the revision.

---

### Official Review · Reviewer_JmYG · 2025-11-01

**Soundness:** 2
**Presentation:** 2
**Contribution:** 2
**Rating:** 6
**Confidence:** 3

**Summary:**

The paper proposes a law of data reconstruction for over-parametrized model in random feature regression showing that the training set can be reconstructed when $p \gg dn$ where, $p$ is parameter count, $d$ is data dimensionality and n the number of samples in the set. They provide elegant theoretical results with empirical validation.

**Strengths:**

* Clear “law” with an intuitive threshold: The paper isolates a memorable scaling for data reconstruction in random-features regression: once the parameter count satisfies ($p \gg dn$), full input recovery becomes feasible. This makes the results feel elegant and easy to understand.
* Crisp geometric criterion (Theorem 1). If the training feature vectors lie in the span of features generated by candidates, each sample must be close (in input space) to some training sample. This spanning-subspace view is conceptually simple and informative.
* Experiments visualize two thresholds: labels fit near ($p\approx n$), while reconstruction error collapses near ($p\approx dn$). The figures make the phenomenon immediately clear.
* Activation analysis. The parity/sign-ambiguity discussion (e.g., ReLU vs. mixed-parity activations) explains when reconstructions may flip signs and how to avoid it, giving a quick diagnostic practitioners can check.

**Weaknesses:**

1. Specialized setting and assumptions: The theory is developed for random-feature regression under sub-Gaussian data, Lipschitz activations, and a strong overparameterized regime (e.g., (p) on the order of (nd) with log factors). This is narrower than many trained deep-net scenarios.
2. Partial completeness of theory. The uniqueness/no-duplicates guarantee is proved cleanly for (n=2); a full characterization of global optima (and uniqueness) for general (n) remains open.

**Questions:**

1. How sensitive is success to initialization and stepsizes? Runs across initalizations, learning rates; report success rates and iterations could help strengthen the paper
2. What is the theoritical challenge in addressing the no duplicates results beyond n=2?
3. How do results change beyond sub-Gaussian isotropic inputs? Statements/experiments under covariance is not $I$ and heavy-tailed features or discussion on how to potentially handle this in the future would help

---

> ### Author Response · Authors · 2025-11-20
>
> We thank the reviewer for the positive evaluation of our work. We address the three questions below. We have edited our paper accordingly and uploaded a revised version, where the changes are marked in red color to make them more visible.
>
> ---
>
> **Question 1**: _Sensitivity of our reconstruction algorithm to initialization and stepsize_
>
> Thank you for the suggestions. Regarding initialization, in all our experiments we report metrics aggregated over multiple distinct random seeds: Figures 3, 6, and 8 report results over 10 seeds, and Figures 1-2 over 4 seeds, with one standard deviation shown as a shaded area, see the corresponding captions. Each seed corresponds to a distinct initialization for $\hat{X}$ and the model weights. We have clarified this point in l. 1776 of the revision.
>
> We note that the optimization of the matrix $\hat X$ occurs on the sphere of radius $\sqrt{d}$. Thus, we found it natural to initialize the rows of $\hat X$ uniformly over such a sphere (since one has to project onto this sphere anyway after the first iteration).
>
> To assess the effect of the step size on the success of the data reconstruction, we have conducted an additional ablation using Binary CIFAR-10 ($n=20$) on the random features model. Starting from the base step size $\eta^* = 2 \cdot 10^3$ (as specified in l. 1713 of the Appendix), we have considered $\eta  \in \\{ \eta^\*/4 , \eta^\*/2, \eta^\*, 2\eta^\*, 4\eta^\* \\}$
> across 10 distinct random seeds. We plot the reconstruction error, the total number of iterations for which the reconstruction algorithm is run and the final reconstruction loss $\mathcal L$, as a function of the number of parameters $p$, in Fig. 11 reported in Appendix E of the revision.
>
> For larger step sizes ($2\eta^\*$ and $4\eta^\*$), we observe oscillatory behavior of the reconstruction loss that prevents convergence within a reasonable budget. Guided by the smallest step size ($\eta^\*/4$), for which the loss consistently converges to zero within at most $6 \cdot 10^4$ iterations across seeds and number of parameters $p$, we cap the optimization at $6 \cdot 10^4$ iterations in this experiment. As shown in Fig. 11, for all step sizes that converge within this budget ($\eta^\*/4, \eta^\*/2, \eta^\*$), the reconstruction error drops to zero once $p \gg dn$, indicating that the reconstruction procedure is robust to the choice of the step size provided it is not taken excessively large. We have clarified this point in l. 511 of the revision.
>
> ---
>
> **Question 2**: _Theoretical challenge in proving the absence of duplicates for $n > 2$._
>
> Our proof technique relies on a contradiction argument. We first assume that the two reconstructed examples $\hat x_1$ and $\hat x_2$ are both approximate copies of the sample $x_1$ (fixed without loss of generality, due to symmetry), see l. 271 of the revised paper. Then, we show that this assumption leads to the contradiction in Equation (10). Now, already when $n=3$, the presence of duplicates can be either given by the `triplet’ $\hat x_1, \hat x_2, \hat x_3$ all similar to each other, or by a pair of duplicates $\hat x_1, \hat x_2$, with a different $\hat x_3$. Our constructive approach would require us to consider these two cases separately, and the amount of cases increases combinatorially for larger values of $n$. This provides the main challenge in extending the proof beyond $n>2$.
>
> Having said that, we note that the key idea of the argument is that duplicate samples ($x_1 + \epsilon_1$ and $x_1 + \epsilon_2$ in the proof of Theorem 2) are close to spanning a degenerate subspace. As a result, our argument consists in proving that the span of the duplicate samples cannot contain higher order terms of the left-out samples ($\tilde \varphi(x_2)$ in the proof of Theorem 2). We suspect that this idea would carry over to a general $n$, as long as $p \gg dn$.
>
> We have clarified this point in the new Remark 2 of the revision, at line 298.

---

> ### Author Response · Authors · 2025-11-20
>
> **Question 3**: _How do results change beyond sub-Gaussian and isotropic inputs_
>
> Thank you for your question. First, we remark that **our work is not restricted to isotropic inputs, but it allows for an arbitrary covariance**, see Assumption 1. We have clarified this point in l. 174 of the revision.
>
> Regarding the assumption of sub-Gaussian inputs, we expect that our results can be extended to a broader set of distributions exhibiting concentration and light tails (see, e.g., the first assumption in (Mei et al., 2022)). Moving to heavy tails, we expect our results to change. For example, the recent work (Adomaityte et al., 2024) considers heavy-tailed data in high-dimensional regression, and it demonstrates a rather different phenomenology with respect to the case with light tails. In the context of our work, a possible research direction would be to investigate if outliers (coming from the heavy tails) can be reconstructed even before the threshold $p \approx dn$, as there is empirical evidence that learning models have a preference towards memorizing outliers (Feldman, 2020). We discuss this point in l. 536 of the revision.
>
> ---
>
> (Mei et al., 2022) Song Mei, Theodor Misiakiewicz, and Andrea Montanari. Generalization error of random feature and kernel methods: Hypercontractivity and kernel matrix concentration. Applied and Computational Harmonic Analysis, 2022.
>
> (Adomaityte et al., 2024) Urte Adomaityte, Leonardo Defilippis, Bruno Loureiro, and Gabriele Sicuro. High-dimensional robust regression under heavy-tailed data: Asymptotics and universality. Journal of Statistical Mechanics: Theory and Experiment, 2024.
>
> (Feldman, 2020) Vitaly Feldman. Does learning require memorization? A short tale about a long tail. In ACM Symposium on Theory of Computing (STOC), 2020.

---

### Author Response · Authors · 2025-11-30

Dear Area Chair,

Thank you for your time and effort in handling our paper. Given the unusual circumstances, we wanted to send a brief message to highlight changes in the revised version following the original reviews.

**1)** The revision contains *additional ablation studies*, including

   - experiments showing the robustness of our reconstruction method for a wide interval of learning rates (Reviewer JmYG);
   - experiments showing that reconstruction is successful if one defines the reconstruction over a higher number of data $\hat n \geq n$ (Reviewer heHJ);
   - successful experiments on vision transformer architectures, TinyImagenet, regularized, and pruned models (Reviewer Krf1, **who replied with a satisfactory feedback**), showing that regularization does not affect the reconstruction method, and that pruning seems to increase the number of parameters needed for data reconstruction.

**2)** Experiments on *classification tasks* (Reviewer hVw8), which give evidence that our proposed law of data reconstruction also appears in classification.

We further thank you and all the Reviewers for their hard work,

Authors

---

### Meta-Review · Area_Chair_D6oB · 2026-01-06

**Summary:**

This paper reveals a law of data reconstruction, showing that the entire training dataset can be recovered once the number of model parameters exceeds the threshold dn, where n is the number of training samples and d is the data dimensionality. Multiple reviewers appreciate the paper and find the results elegant and easy to understand, the geometric criterion crisp, and the theoretical and empirical findings convincing.

**Reviewer Concerns:**

Reviewers raised several concerns, such as the analysis being restricted to random features and small networks, the experiments relying on small sample sizes (mainly 32\times 32), and questions about transferability to transformers. The AC finds that the authors have provided convincing responses to most of these concerns.

**Reviewer Scores:**

This paper receives the following ratings: Marginally Above, Accept, Marginally Above, and Marginally Above. If the reviewers had been able to participate fully in the discussion, the AC would expect these positive ratings to be maintained or increased. Thus, the AC recommends accepting this paper.

---

### Decision · Program_Chairs · 2026-01-26

Accept (Poster)